# How machine learning can help select capping layers to suppress perovskite degradation

Noor Titan Putri Hartono [1], Janak Thapa[1], Armi Tiihonen [1], Felipe Oviedo[1], Clio Batali[1], Jason J. Yoo[1], Zhe Liu[1], Ruipeng Li[2], David Fuertes Marrón[1,3], Moungi G. Bawendi [1], Tonio Buonassisi [1✉] & Shijing Sun [1✉]

Environmental stability of perovskite solar cells (PSCs) has been improved by trial-and-error exploration of thin low-dimensional (LD) perovskite deposited on top of the perovskite absorber, called the capping layer. In this study, a machine-learning framework is presented to optimize this layer. We featurize 21 organic halide salts, apply them as capping layers onto methylammonium lead iodide (MAPbI$_3$) films, age them under accelerated conditions, and determine features governing stability using supervised machine learning and Shapley values. We find that organic molecules' low number of hydrogen-bonding donors and small topological polar surface area correlate with increased MAPbI$_3$ film stability. The top performing organic halide, phenyltriethylammonium iodide (PTEAI), successfully extends the MAPbI$_3$ stability lifetime by 4 ± 2 times over bare MAPbI$_3$ and 1.3 ± 0.3 times over state-of-the-art octylammonium bromide (OABr). Through characterization, we find that this capping layer stabilizes the photoactive layer by changing the surface chemistry and suppressing methylammonium loss.

[1] Massachusetts Institute of Technology, 77 Massachusetts Avenue, Cambridge, MA 02139, USA. [2] National Synchrotron Light Source II, Brookhaven National Laboratory, Upton, NY 11973, USA. [3] Instituto de Energía Solar-ETSIT, Universidad Politécnica de Madrid, 28040 Madrid, Spain. ✉email: buonassisi@mit.edu; shijings@mit.edu

Perovskite solar cell (PSC) stability is still far less than the ~25 years required to enter the mainstream photovoltaic (PV) market[1], despite efficiencies reaching 25.2%[2]. Improving environmental stability is a critical step. Recent studies suggest that mixing low-dimensional (LD) perovskite with the absorber improves the stability, but device performance suffers because carrier transport is reduced due limited carrier mobility through the LD material[3–5].

In contrast, the capping-layer method improves poor carrier transport by intercalating the 2D perovskite with conductive organic materials. As a result, short-circuit current ($J_{SC}$) recovers. With improved surface passivation because of the capping layer, the open-circuit voltage ($V_{OC}$) increases, as does environmental stability at ambient temperature with elevated (40–90%) relative humidity (RH)[6,7]. The capping layer is formed by reacting organic halides in a solvent with the 3D perovskite network underneath, forming the LD perovskite network with intercalated organics. The choice of organic halides is known to affect device stability, however the relationship between structure and stability has not been fully explored, in part because the parameter space is vast[8]. Perovskite thin-film deposition with a range of organic halides has been reported, including benzene rings/phenyl with amine (e.g., phenylethylammonium iodide[9–11], phenylethylammonium bromide[9,12], aniline iodide[13,14], benzylammonium iodide[14], teophylline[15], caffeine[16]), long carbon chains with amine (e.g., n-butylammonium iodide[17,18], n-octylammonium iodide[18]), fluorous amine (e.g., 2-(4-fluorophenyl)ethylammonium iodide[19]), branched amines (e.g., 1,8-octanediammonium iodide[20], diethylammonium bromide[21], diethylammonium iodide[21], n-hexyl trimethyl ammonium bromide[6]), and large, complex structures (e.g., Eu-porphyrin complex[22]). Photovoltaic devices based on these materials demonstrated improved stability and efficiency than their non-capped controls under various environmental test conditions. However, little is known about which specific chemical properties among the different types of organic halides control the improved stability of the capped materials.

Inspired by recent studies on inverse design of polymers and inorganic solids[23–25], as well as on using machine learning to understand PSCs' properties[26–28], we present a machine-learning framework to investigate LD organic-inorganic perovskites serving as a capping layer for MAPbI₃. We elucidate which properties of capping layers are responsible for enhancing stability, and the underlying mechanisms whereby they work. With this information, we can generate materials-design guidelines.

## Results

**Study overview and objectives**. We consider 21 organic salts as capping-layer materials, with different sizes, branches, and chemical properties, including both iodine and bromine-based salts. Capping layers are deposited using spin coating atop 300 nm thick films of methylammonium lead iodide ($CH_3NH_3PbI_3$, or MAPbI₃)[29]. The poor MAPbI₃ stability guarantees a strong baseline degradation rate, and strong signal-to-noise for our study. (In principle, the framework developed in this study can be extended to different perovskite absorber compositions, including mixed-cation and -anion materials that gained popularity in recent years.) For each film, 12 different processing conditions are explored. Following sample fabrication, perovskite films are tested unencapsulated under rigorous accelerated aging conditions (85% RH, 85 °C temperature, and 0.16 Sun illumination). We photograph the samples in situ every 3 min, calibrate color using calibration tiles with thin-plate spline color warping method[30], and extract numerical values for degradation onset and rate from the time series images as a proxy for film stability[28,31–33].

To determine which capping-layer properties and processing conditions govern film stability, we employ a supervised-learning algorithm with a feature importance ranking. As model inputs, we include structural and chemical features of the organic molecules in the capping layers, derived from the PubChem 2019[34] database, as well as 12 processing conditions. The 12 processing conditions vary capping-layer annealing temperature and capping-layer precursor solution concentration. As model output, we use degradation onset and rate. We then determine the feature importance ranking, using Shapley value concept[35], and use this ranking to infer design rules for organic molecules comprising capping layers.

The model trained on our experimental data, and subsequent feature importance ranking, indicate that the number of hydrogen-bond donors and the organic-molecule topological surface area are the two most important features of an organic capping-layer molecule governing film stability. To determine why our best-performing molecule exhibits the best stability among the 21 screened materials, we perform in-depth materials characterization, examining both the surface and the bulk.

Figure 1a shows the overview of the study and objectives for finding the design guidelines of capping layer for suppressing degradation in perovskite solar cells. Figure 1b shows the example of average change in color in the accelerated aging test, from black to yellow.

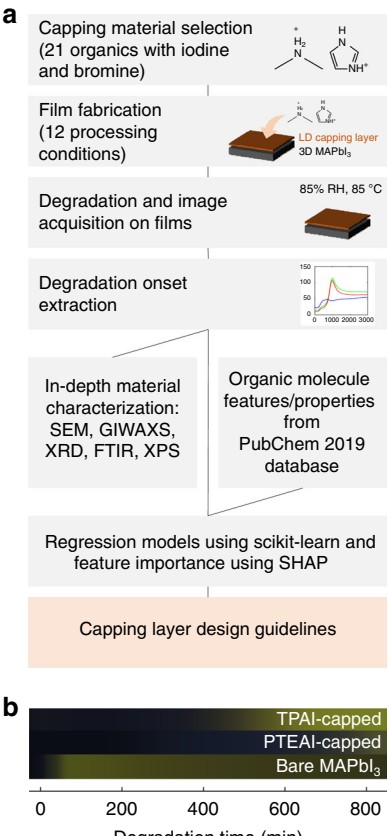

**Fig. 1 The workflow of the study and aging test result over time.**
**a** Schematic overview of this study, aimed at developing design rules for capping layer of perovskite solar cells. **b** the raw image changes for tetrapropylammonium iodide (TPAI)-capped, phenyltriethylammonium iodide (PTEAI)-capped, which have similar molecular weights, and bare MAPbI₃ films.

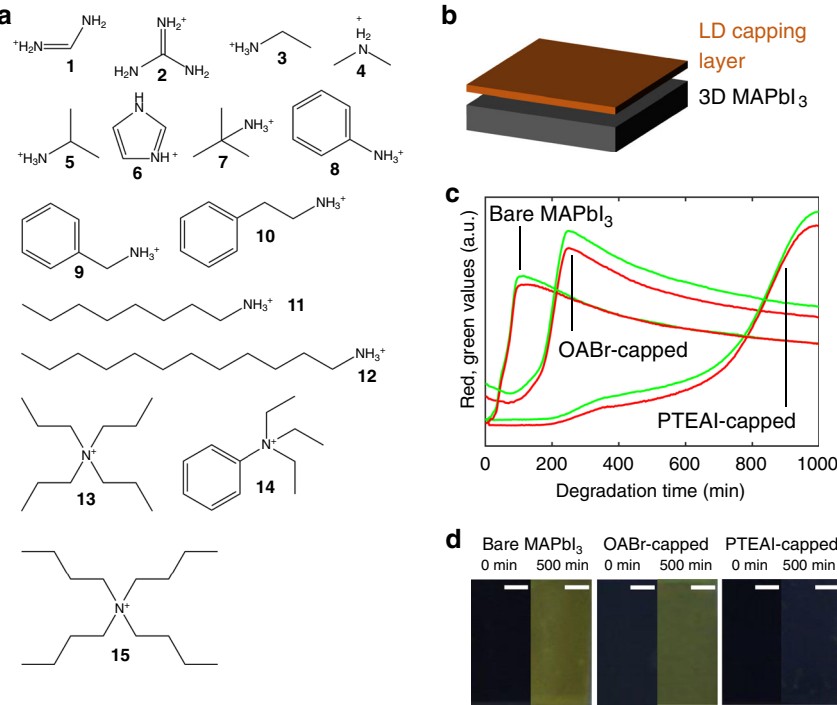

**Fig. 2 Capping-layer precursors and accelerated aging results of best-performing capping-layer material. a** *A*-site cations used in this study and their chemical structure: (1) formamidinium (FA), (2) guanidinium (GA), (3) ethylammonium (EA), (4) dimethylammonium (DMA), (5) *iso*-propylammonium (iPA), (6) imidazolium (ID), (7) *t*-butylammonium (tBA), (8) phenylammonium (PhA), (9) benzylammonium (BzA), (10) phenethylammonium (PEA), (11) *n*-octylammonium (OA), (12) dodecylammonium (DA), (13) tetrapropylammonium (TPA), (14) phenyltriethylammonium (PTEA), and (15) tetrabutylammonium (TBA). **b** The film structure with 3D MAPbI$_3$ at the bottom and LD capping layer deposited atop. **c** The time-dependent red and green values of camera images, and **d** the camera images of pre-degraded and 500 min-degraded films for bare MAPbI$_3$ control material, (11) OABr, used in state-of-the-art high-efficiency devices, and (14) PTEAI, our best-performing material in this study, with scale bar 0.5 cm.

**Capping-layer composition, fabrication, and aging tests.** Capping-layer precursors consist of 15 *A*-site organic cations in different lengths and shapes (Fig. 2a). The number of carbon atoms in these materials ranges between 1 and 16, with primary (two N–H bonds), secondary (one N–H bond), tertiary (no N–H bond) amine, or quaternary (no N–H bond, no lone pairs). Two *X*-site anions, iodide and bromide, are tested. A total of 21 unique organic-halide salts are explored. The *AX* capping-layer material is dissolved in solvent, spin coated atop a MAPbI$_3$ thin film with excess PbI$_2$[36] (Fig. 2b), and annealed at temperatures between 50 and 125 °C for 10 min.

For the samples that stay dark for longer than 300 min, each sample condition is repeated for at least two times to ensure adequate statistics. The samples were aged under 85% RH, 85 °C temperature, and 0.16 Sun visible-only illumination in batches of 28 samples. Each aging test is stopped after the sample turns yellow, indicating the degradation from perovskite black phase into the PbI$_2$ phase[37,38]. Illumination (0.16 Sun) is added, which allows optical images to be taken every 3 min as a proxy for film degradation. The red, green, and blue (RGB) color values are extracted for different time points of the degradation from the image data. The increasing red and green (RG) colors correspond to changes in film color from black to yellow, as shown in Fig. 2c, d. The onsets are described as the rapid change from black to yellow evidenced in the red and green (RG) channels (Supplementary Figs. 1 and 2), which are the key descriptors for stability in this study. Although this study focuses on the degradation onset, it is possible to consider the rate of color change (or the slope, Supplementary Fig. 3). Because both red and green channels overlap, it is sufficient to consider only the red channel (Supplementary Fig. 4). The complete comparison of film color before and during the degradation is shown in Supplementary

Fig. 2, where the bare MAPbI$_3$ color change onset occurred at 107 min on average, and MAPbI$_3$ films with specific capping layers, e.g., tetrapropylammonium iodide (TPAI), tetrapropylammonium bromide (TPABr), tetrabutylammonium iodide (TBAI), tetrabutylammonium bromide (TBABr), and phenyltriethylammonium iodide (PTEAI) retained their dark color 4 ± 2 times longer. The longer alkyl chain performs better than the shorter one. Branched molecule and phenyl group molecule also lead to better film stability. All the most stable capping-layer materials in this study have quaternary ammonium group, instead of primary, secondary, or tertiary. The quaternary ammonium has been shown to effectively passivate the charged defects and help to minimize the initiation of film degradation[39].

**Machine-learning regression, feature importance ranking, and design principles.** Machine-learning regression is performed on a color change-based degradation descriptor described in the previous section. Colors of the samples are extracted from JPEG pictures that have been color calibrated to ensure reproducibility and repeatability. Specifically, the onsets, i.e., time-intercepts of rapid color change from black to yellow, are the key descriptors for stability in this study. The onset time is a continuous variable; hence we are using regression as supposed to classification machine-learning models. This onset descriptor acts as the output of our machine-learning model, which is used to train the models with the input coming from the database and the processing condition.

We featurize 21 organic capping-layer materials using their material properties from the PubChem 2019 database[34], namely molecular weight, *x* log *P* (or partition coefficient that indicates hydrophobicity/hydrophilicity of molecules)[40], the number of

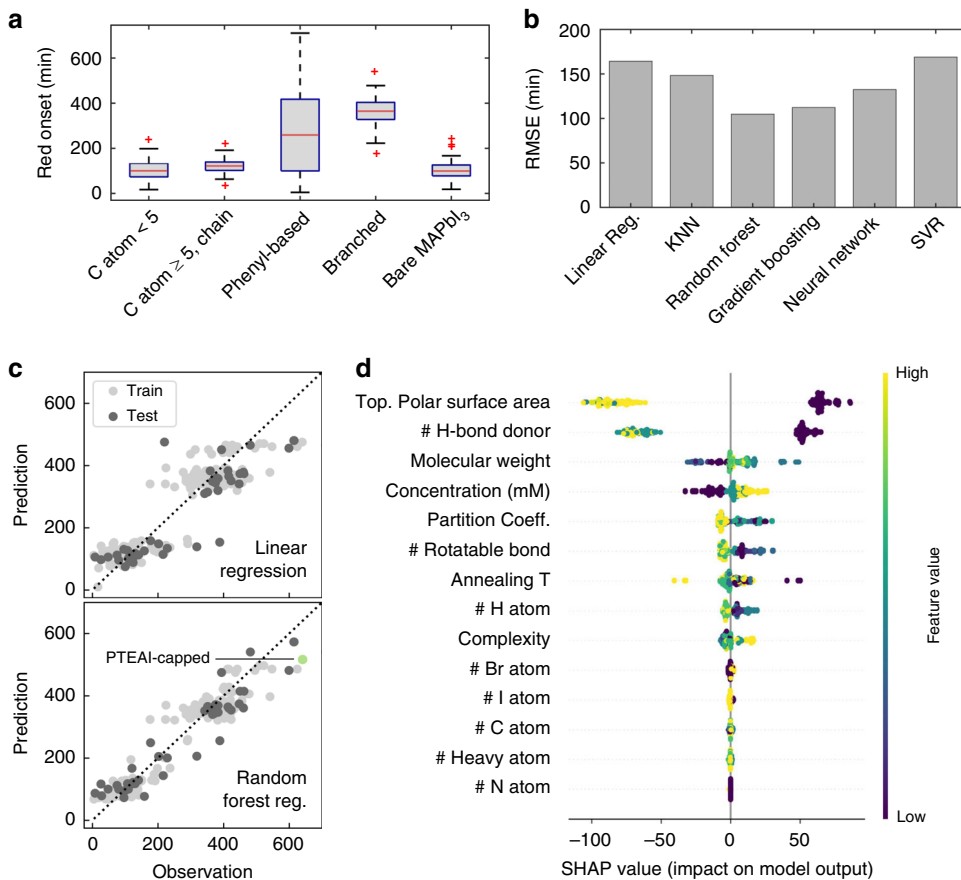

**Fig. 3 Image data parameter extraction, regression results, and Shapley-based importance rank. a** Extracted red onset points (in minutes) and their distributions from time-dependent camera data of capping-layer groups and bare MAPbI$_3$ controls, synthesized using 12 different processing conditions. The red line shows the median, and the box shows the upper and lower quartiles. **b** The cross-validated root mean square error (RMSE) of various machine-learning models, including linear regression, K-nearest neighbors (KNN), random forest regression, gradient boosting regression with decision trees, neural network (multilayer perceptron), and support vector regression (SVR), with non-normalized input. **c** The comparison of red onset prediction and observation output between linear regression and random forest regression, with the green circle indicating PTEAI-capped film. **d** The feature importance ranking obtained from the random forest regression algorithm and SHAP library, showing the chemical properties and processing conditions in descending order of importance (rank). The yellow and purple color indicates high and low values of a given feature, respectively.

rotatable bonds, complexity[41,42], topological polar surface area (TPSA)[43], the number of hydrogen-bond donors[44], and the number of each element (carbon, hydrogen, bromine, nitrogen, and iodine). For each of the 21 candidate materials, we test 12 unique processing conditions by varying the annealing temperature of capping layer after spin coating step, and the concentration of capping-layer precursor solution. These material-property and processing-condition features are used as inputs into our machine-learning models, and are described in more detail in Supplementary Table 1. The 21 capping-layer materials can be grouped arbitrarily into four different groups: organics with carbon atom fewer than 5, long-chain organics with carbon atom more than 5, phenyl-based organics, and branched organics, as shown in Fig. 3a. In general, both phenyl-based[14] and branched[45] organics have previously been reported to increase stability.

Using the scikit-learn library in Python[46], 6 regression models are trained on the data from 260 accelerated aging tests shown in Fig. 3a, including linear regression, K-nearest neighbors (KNN) regression[47], random forest regression[48], gradient boosting regression with decision trees[49], multilayer perceptron neural network with three hidden layers (each has 128, 256, and 64 units) by using Adam solver[50], and support vector machine regression (SVR)[51]. The hyperparameters in different models are optimized (Supplementary Tables 2 and 3) using GridSearchCV

function on scikit-learn library, which performs an exhaustive search of parameters based on minimum fivefold cross-validated root mean squared error (RMSE). The ML input is either normalized, which calibrates the mean to zero and scales to unit variance, or not. The fivefold cross-validated RMSE result for both input types are shown in Supplementary Fig. 5. Random forest regression results in the lowest fivefold cross-validated RMSE for non-normalized input, as shown in Fig. 3b. Random forest regression is a method with several decision trees, where each of their estimators is independently predicted from a different subset of data, and in the end, the estimators are averaged[52]. The RMSE of linear regression is quite high and it has inconsistent weights (as shown in Supplementary Figs. 6 and 7). The RMSE of multilayer perceptron neural network is large due to dataset that is small to be used with neural-network method. RMSE's of the random forest regression and gradient boosting regression with decision trees are lower, about 104 and 112 min, respectively, which is still high, considering the degradation onset range of 0–700 min. The high RMSE is caused by the variability in 12 different synthesis conditions, in addition to inherent high variability in the bare MAPbI$_3$'s degradation profile[53,54] (standard deviation of red onset across 35 bare MAPbI$_3$ samples ≈ 45 min), as shown in Fig. 3a and Supplementary Table 4. Figure 3c demonstrates the randomly-split 20%:80% test:train set

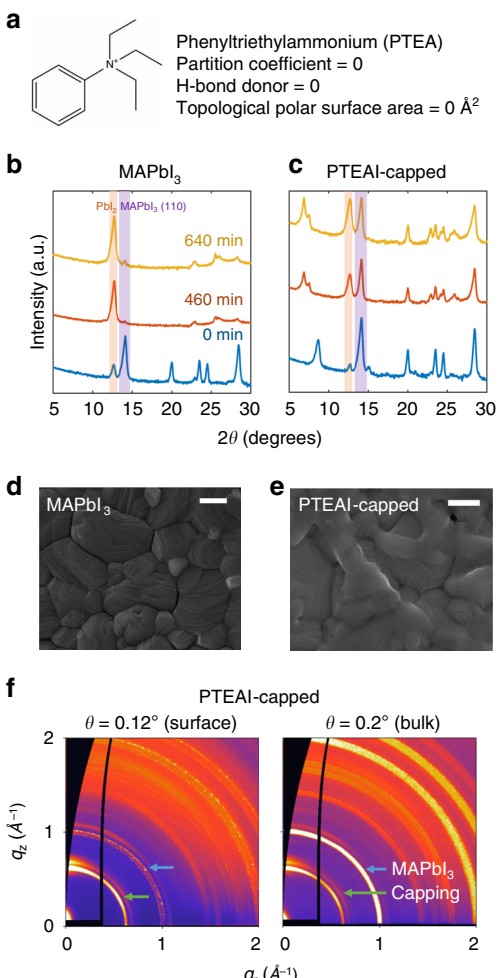

**Fig. 4 Crystal structure and morphology characterization for optimum capping layer (PTEAI). a** Phenyltriethylammonium (PTEA) has 0 hydrogen-bond donor due to its quaternary amine. Powder X-ray diffraction (XRD) spectra of bare (**b**) and PTEAI-capped MAPbI₃ (**c**), when they were degraded for 460 and 640 min The purple shaded area indicates the (110) peak of MAPbI₃, and the orange shaded area indicates the (001) PbI₂ peak. Pre-degraded scanning electron microscopy (SEM) image of bare (**d**) and PTEAI-capped MAPbI₃ (**e**), scale bar: 200 nm. **f** Grazing-incidence wide-angle x-ray scattering (GIWAXS) images of PTEAI-capped MAPbI₃ presenting the structure on the surface and in the bulk films, with the incident angles of 0.12° and 0.2°, respectively. The green arrow points to LD capping peak, and the blue arrow points to the MAPbI₃ peak.

observation and prediction results of random forest regression and linear regression, which shows that the random forest model is the best-performing model for our data, without the evidence of overfitting.

To interpret our model, we further analyze the random forest regression results using SHAP (Shapley Additive exPlanations)[35], a generalized metric for feature importance, which utilizes the game-theory-based Shapley values to calculate the contribution of each feature to the model's output. SHAP indicates how each feature contributes to the model output (red color change onset time). Figure 3d shows the 14 features (inputs) ranked using SHAP based on random forest regression. The yellow color corresponds to high value of the features consisting of molecular properties and processing conditions, whereas the purple color corresponds to low value of the features. The x-axis of Fig. 3d, labeled as the SHAP value (impact on the model output), represents the red degradation onset values. If the SHAP value is

positive, the degradation onset increases hence the film lasts longer, and vice versa. The y-axis of Fig. 3d, listing all the 14 features, are ranked based on their contribution to the degradation onset.

Low number of hydrogen-bond donor and small topological polar surface area (TPSA) are the top most important factors determining stability, shown as the yellow points on the positive side of SHAP value. PTEAI, as the most stable capping-layer material in this study, indeed has a low number of hydrogen-bond donor (0) and a small TPSA (0 Å²). The statistical analysis (ANCOVA) of the films shows that the red degradation onset of the most stable capping-layer material in this study, PTEAI, is statistically significantly different, with 95% confidence level, in comparison to other materials and bare MAPbI₃ film (Supplementary Fig. 8). This result is also consistent with gradient boosting regression with decision trees method (Supplementary Fig. 9). Both number of hydrogen-bond donor and TPSA are correlated, because the hydrogen-bond donor presents when there is a bond between electronegative atoms (in our case, nitrogen) and hydrogen, creating a polar surface area on the molecule. The Pearson correlation value coefficient of hydrogen-bond donor and TPSA is 0.81 (Supplementary Fig. 10)[43]. This evidence might support the hypothesis that hydrogen bonding plays a very important role in degradation of perovskite solar cells[55–59], especially under high-humidity testing conditions. The next important features which affect stability are molecular weight and concentration of precursor solution. On the other hand, x log P or partition coefficient (an indicator for hydrophobicity), complexity, and the number of carbons, iodines, or bromines generally have lower ranks in the model. If we consider the recently published capping-layer materials, such as theophylline, caffeine, and theobromine[15,16], assuming they were fabricated and aged in the same manner as the materials in this study, the onsets are predicted to happen at 103.2, 264.2, and 121.5 min respectively. As a reference, PTEAI onset happens at (462 ± 115) min. Due to its lower number of hydrogen-bond donor and smaller polar surface area, caffeine is predicted to be more stable among these three materials, even though its surface passivation property is worse than theophylline[15]. If the topological polar surface area and number of hydrogen bonding are indeed the most important features, in future studies researchers can explore more complex quaternary ammonium group ($NR_4^+$), where R is an alkyl or an aryl group, for instance, N,N,N-trimethylnaphthalen-1-aminium iodide (Supplementary Fig. 11).

The advantage of using SHAP instead of traditional interpretability methods, is its robustness to correlated features[35] than traditional interpretability methods, due to the game-theoretic nature of feature attribution. However, if strong multicollinearity exists (i.e., very high shared variance among features causing significant variations in model training), highly correlated features, for instance the molecular weight and the number of carbon atoms, might be incorrectly attributed feature importance, according to how the model is fitting the data. To diminish these spurious attributions, we train various different models via cross-validation and make our final decision based on contrasting the feature ranks (Supplementary Figs. 12–18). The consistency among models, along with testing using cross-validation, allows us to physically interpret the models. The most robust approach, which is incompatible with our exploratory goals, could have been to perform causal inference using randomized control trials or experiments in a more limited compound space. The tradeoff, of course, is the slow and poor exploration of the material space.

**Protection mechanisms in top-performing candidates.** As suggested by feature importance ranking, our top-performing capping-layer material, PTEAI, which is reported to have no

hydrogen bonding acceptor and 0 Å² of TPSA as shown in Fig. 4a, does achieve high stability[34]. The PTEAI-capped perovskite enables 1.3 ± 0.3 times and 4 ± 2 times improvement in stability, to the state-of-the-art OABr capping layer that we test in this study, and bare MAPbI₃ films, respectively[7]. To better understand the protection mechanisms achieved by the PTEAI capping layer, we further compared PTEAI-capped perovskite with other capping layers such as TPAI, which contains the same number of carbon atoms as PTEAI and has a similar molecular weights but degrades faster, via X-ray diffraction (XRD), scanning electron microscopy (SEM), and grazing-incidence wide-angle-X-ray scattering (GIWAXS), as shown in Supplementary Figs. 19–26.

The powder XRD data are compared to butylammonium (BA)-based 2D perovskite to confirm the new phase[60-62]. The addition of a capping layer indeed introduces a new phase in the film, indicated by the emergence of a new peak at 8.68° in the case of a PTEAI capping layer, which matches with $(BA)_2(MA)_3Pb_4I_{13}$, as shown in Fig. 4c and Supplementary Fig. 19. This indicates a Ruddlesden–Popper (RP) perovskite phase, $(PTEA)_2(MA)_3Pb_4I_{13}$ ($n = 4$)[60-62]. $n$ indicates the number of 3D perovskite layers before separated by organic molecules, and the RP perovskite formula is in the form of $A'_2A_{n-1}B_nX_{3n+1}$, where $A$, $B$, and $X$ correspond to $A$-site cation, $B$-site cation, and $X$-anion. In the case of the TPAI capping layer, the new peak matches with $(BA)_2(MA)_2Pb_3I_{10}$, indicating a RP perovskite phase, $(TPA)_2(MA)_2Pb_3I_{10}$ ($n = 3$), as shown in Supplementary Fig. 20[60-62]. The capping layer reacts with the excess $PbI_2$ coming from the MAPbI₃ layer underneath, forming the RP perovskite phase at the top[7].

Within 460 min of accelerated aging tests, almost all of the bare MAPbI₃ degrades into lead iodide ($PbI_2$), as indicated by the emergent peak at 12.64° and almost complete suppression of the MAPbI₃ related peak at 14.02°, highlighted by the shaded purple area in Fig. 4b. The PTEAI-capped films, on the other hand, maintains its MAPbI₃ and RP peaks (Fig. 4c), albeit shifted due to thermally induced structural modification[63], much longer than bare or TPAI-capped films, shown in Supplementary Fig. 21. This is further evidence that the RP phase in the capping layer reduces the MAPbI₃ degradation rate, because it helps suppress the conversion of MAPbI₃ into $PbI_2$.

Comparing the surface morphology of MAPbI₃ and PTEAI-capped MAPbI₃, we noticed a difference at pre-degradation time point, where the capping layers coat the surface of MAPbI₃, including the grain boundaries, as shown in Fig. 4d, e. As degradation occurs, and the surface reacts with the high-humidity environment at elevated temperature, the grains change and more pinholes appear, as shown in Supplementary Fig. 22.

GIWAXS images provide a deeper understanding about the crystal structure difference of the surface with respect to the bulk of the films. Figure 4f shows the pre-degraded data for the surface at low incidence angle ($\theta = 0.12°$) below the critical angle of perovskites (0.18°), and the bulk at higher incidence angle ($\theta = 0.2°$) along $q_r$ (horizontal) and $q_z$ (vertical) axes. The blue arrow for capped samples shows the signature of capping materials/RP phase, whereas the green arrow shows the MAPbI₃ phase. The ratio of the RP phase of PTEAI capping layer and the MAPbI₃ phase close to the surface is much larger than deeper in the bulk, which is dominated by the MAPbI₃ phase. This indicates that the capping-layer material mostly resides on the surface of the perovskite thin film. In addition, we find that the LD perovskite based on PTEAI and TPAI show different texture, based on their vertical and horizontal integration of GIWAXS data. Further analysis on crystallite textures from GIWAXS is shown in Supplementary Figs. 23–26.

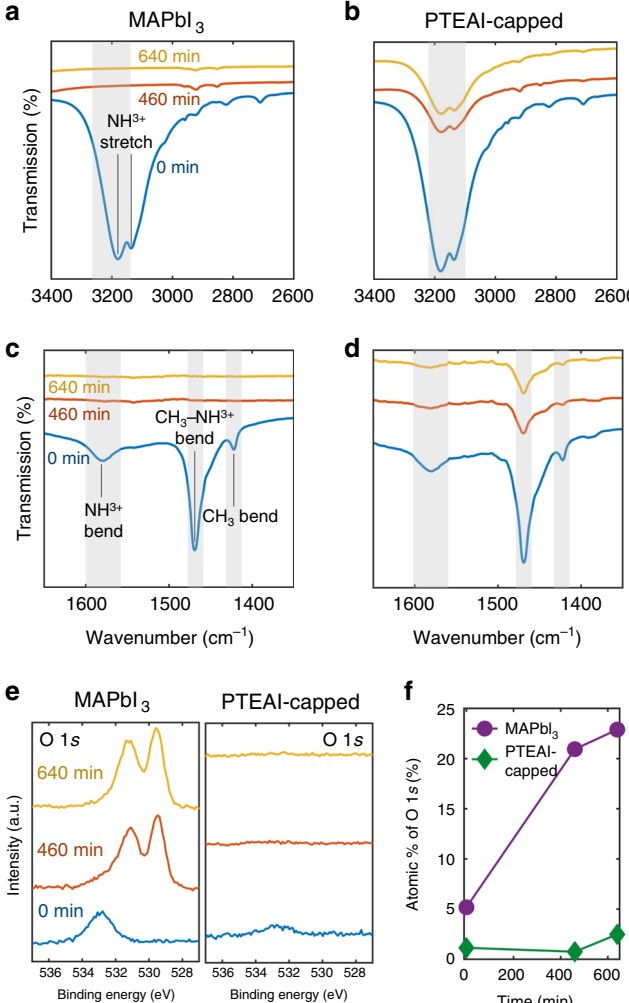

**Fig. 5 Capping-layer surface chemistry characterization of optimum capping layer (PTEAI).** Attenuated total reflection Fourier-transform infrared spectroscopy (ATR-FTIR) with ZnSe crystal of bare (**a**, **c**) and PTEAI-capped MAPbI₃ (**b**, **d**). **e** The O 1s spectra obtained from x-ray photoelectron spectroscopy (XPS) of bare and PTEAI-capped MAPbI₃, and **f** their corresponding atomic percentages. The purple circle and the green diamond represent the atomic percentages of O 1s for bare and PTEAI-capped MAPbI₃ film, respectively.

The feature importance ranking indicates hydrogen bonding donor and topological polar surface area as the most important capping-layer features, toward which the amine in the organics significantly contributes. We hence investigate the surface chemistry of the organic molecules in PTEAI-capped films. Changes in organic-molecule bonds within the perovskite films can be detected using Fourier-transform infrared spectroscopy (FTIR) attenuated total reflection (ATR) geometry with a zinc selenide (ZnSe) crystal. We measure bare MAPbI₃ and PTEAI-capped MAPbI₃ using FTIR, and observe the main signature of methylammonium (MA) at 3176 and 1580 cm⁻¹, indicated by NH³⁺ stretch (Fig. 5a, b) and bend (Figs. 5c, d) respectively. This result suggests that MA in bare MAPbI₃ disappeared at 460 min, whereas MA in the PTEAI-capped film remained after 460 mins.

Using X-ray photoelectron spectroscopy (XPS), we observed no significant traces of oxygen (O 1s) on the surface of PTEAI-capped film, even after 640 min of accelerated aging tests (Fig. 5e), revealing that the PTEAI-capped MAPbI₃ inhibits the formation of an oxygen-containing compounds[64], and increases film

stability. In contrast, we found that the amount of oxygen-containing compounds[64] in bare MAPbI₃ increases within 460 min, which can be attributed to a $PbO_x$ compound[64,65]. The presence of the oxygen in fresh bare MAPbI₃ indicating surface contamination from atmospheric oxygen is significantly higher than in fresh PTEAI-capped samples. After fitting the C 1s, I 3d, N 1s, O 1s, and Pb 4f XPS peaks, the atomic percentages of each element are calculated; for MAPbI₃, the O 1s atomic composition percentage increases from about 5–20% after aging for 460 min, whereas for PTEAI-capped MAPbI₃, the O 1s atomic composition percentage stays below 1%, as shown in Fig. 5f.

Considering the inorganic materials change on the film surface, the XPS scans of Pb 4f reveal the presence of two different Pb-containing species, in the near-surface region of the capped layers, indicated by two doublet peaks, as shown in Supplementary Fig. 27 and in agreement with the formation of a second Pb-containing species in the RP phases of both TPAI- and PTEAI-capped samples. In bare MAPbI₃, the initial one doublet peak of Pb 4f is at 138.8 eV, and shifts to 138.2 eV as it degrades. The capping layers, on the other hand, initially has two doublet peaks, indicating the presence of two different Pb-bonds on the surface. After 640 min, PTEAI-capped films still preserve its two doublet peaks, while the TPAI-capped film's extra peak disappears. The co-existence of the two distinct Pb-doublets in capping-layer samples as a function of time can therefore be directly correlated with the resilience of the capping-layer surface to air exposure and the observed stability of the absorber material.

## Discussion

In this study, we present a machine-learning-assisted investigation of the features that increase the effectiveness of hybrid organic-LD perovskite capping layers atop lead-halide perovskite solar cells. We test MAPbI₃ coated with 21 combinations of organic molecules and halide anions, under accelerated aging conditions of 85% RH, 85 °C, and 0.16 Sun illumination, and featurize the organic molecules according to open-source database values. We apply a random forest regression algorithm and SHAP values to identify which features correlate most strongly with improved stability, and find that the most important properties extending the degradation onset are (i) the low number of hydrogen-bond donors and (ii) the small topological polar surface area of the organic capping-layer molecules. By utilizing the organic salt that exhibits the strongest features, phenyl-triethylammonium iodide (PTEAI), we increase the stability of bare MAPbI₃ and state-of-the-art high-efficiency MAPbI₃ with an OABr-based capping layer by more than $4 \pm 2$ and $1.3 \pm 0.3$ times, respectively. Synchrotron-based XRD indicates a new Ruddlesden–Popper perovskite, $(PTEA)_2(MA)_3Pb_4I_{13}$, which serves as a capping layer. XPS and FTIR reveal that the top-performing capping layer stabilizes the MAPbI₃ perovskite by modifying the surface structure and chemistry, which coincides with suppression in the methylammonium loss and formation of both PbI₂ and oxygen-containing compounds at the surface of perovskite. Our findings suggest capping-layer design rules that may further enhance the environmental stability of halide perovskite-based devices under harsh conditions, and pushing perovskite-based solar cells closer toward mainstream photovoltaics market.

## Methods

### Film and capping-layer fabrication
For 3D methylammonium lead iodide (MAPbI₃) precursor solution: 1.5 M PbI₂ (TCI Chemicals) solution was dissolved in 9:1 DMF:DMSO mixed solvents, before mixing them with ammonium powder. For every gram of methylammonium iodide (MAI) powder (Dyenamo), we added 5.10 mL PbI₂ stock solution correspondingly, which corresponds to MAI:PbI₂ molar ratio of 1:1.09. Capping-layer solutions were made in three different concentrations, 5, 10, and 15 mM, by mixing ammonium iodide/ammonium bromide powder with isopropyl alcohol, pure, ACS reagent, ≥99.5% (Sigma-Aldrich). A list of ammonium iodide/ammonium bromide powder manufacturers is listed in Supplementary Table 5.

65 µL of MAPbI₃ solution was then deposited on the precleaned substrate (glass slides for UV-Vis, GIWAXS, and FTIR and XRD, FTO substrates for SEM, and XPS), and spin coated with this 2-step recipe: 1000 rpm for 10 s and acceleration of 200 rpm/s, then 6000 rpm for 30 s and acceleration of 2000 rpm/s. 5 s after the start of the second step, 150 µL of chlorobenzene was dropped on the substrate. Then, the deposited film was annealed on the hotplate at 100 °C for 10 min. After the substrate is cooled down, 60 µL of capping-layer solution was deposited on top, and spin coated with 3000 rpm speed for 30 s. The substrate was then annealed with various temperatures, 50, 75, 100, and 125 °C, for 10 min.

### General characterization
The crystal structure and the film phases were characterized using X-ray diffraction (XRD, Rigaku SmartLab), with Cu-Kα sources. The film morphology and device cross-section were investigated using a ZEISS Ultra-55 field-emission scanning electron microscope (FESEM, ZEISS). The X-ray photoelectron spectroscopy was measured using K-Alpha+ XPS (Thermo Scientific) with Al-Kα excitation source. The Fourier-transform infrared spectroscopy (FTIR) was measured using Perkin-Elmer Spectrum 400 (Perkin-Elmer), in attenuated total reflection (ATR) configuration with ZnSe and Ge crystals. Samples were stored in inert conditions inside a nitrogen-purged glovebox between synthesis and aging test/characterization steps.

### GIWAXS characterization
Grazing-incidence wide-angle X-ray scattering (GIWAXS) measurements on perovskite thin films were taken at beamline 11-BM (CMS) at the National Synchrotron Light Source II (NSLS-II) of Brookhaven National Laboratory. The X-ray beam with the energy of 13.5 keV shone on the thin films in the grazing incident geometry. Multiple incident angles were chosen to tune the X-ray penetration of the films and probe the structure of the surface and the bulk. The scattering spectra were collected with the exposure time of 30 s by an area detector (DECTRIS Pilatus 800 K) placed 257 mm away from the sample. The data analysis was performed by using custom-made software SciAnalysis[66].

### Accelerated aging chamber and image acquisition
The images during accelerated aging tests were acquired using a Thorlabs DCC1645C CMOS USB camera (with IR-Cut-Filter 650 nm removed), taken every 3 min automatically using LabView software. X-Rite Color Checker Passport 2 was used as a color reference to transform the sample images to the L*a*b color space. The color calibration used 3D thin-plate spline color warping method, and the resulting data were transformed back to red, green, blue (RGB) color space[30]. JPEG is a more compressed file format than raw bitmap BMP. The quantitative RG color values extracted from both formats, from the initial black perovskite phase until they completely degrade and turn yellow, show negligible difference (Supplementary Fig. 28). As the JPEG image data are more compressed and hence are faster to be processed, are used as the stability proxy (Fig. 2c; Supplementary Fig. 2). The relative humidity (RH) set point was maintained at $85 \pm 3\%$ using an Arduino-controlled feedback system, and both the RH and temperature were measured using an Adafruit Si7021 sensor and EasyLog EL-USB-2 data logger. The visible only white illumination intensity in the chamber was 0.16 Sun, using an Advanced Illumination DL097 LED lamp. The samples were heated using in-house-built graphite heating elements, controlled at $85 \pm 2$ °C. Sample placement inside the aging chamber was randomized, to minimize risk of systematic placement-related errors. The humidity, heating element temperature, and chamber temperature were recorded throughout the test, ensuring the environment humidity and temperature profiles in each round were the same.

### Data integrity
The synthesis conditions were recorded by the experimenter using a laboratory notebook, then transcribed to Google Sheets. Accelerated aging test data (camera image time series) were automatically pushed to Dropbox, and subsequently quantified using the 3D thin-plate spline color warping method. Features were extracted from calibrated RGB data using Python and MATLAB, where various parameters including red-channel onset (the time-intercept of red color degradation that corresponds to the yellowing/changing of perovskite phase into PbI₂) were extracted onto a local computer. Raw GIWAXS, XPS, and FTIR data were processed using their own software packages, with different file labeling conventions, and stored on different local computers. Metadata (linking different files containing synthesis conditions, calibrated aging test data, GIWAXS, XPS, and FTIR data) were created on an ad-hoc basis, as samples were deemed of high scientific significance. Not all the data obtained are reported in this paper.

### Machine-learning analysis
All the features' values are numerical and can be treated as continuous variables, which therefore do not need further encoding. The capping-layer material name is not used as one of the features, because its properties have been included in the features instead. Therefore, the preprocessing done is normalization of the input data for the models. We performed normalization of the model inputs ($X$) using the *StandardScaler* algorithm, in the scikit-learn library[46], which calibrates the mean and scales to unit variance. The inputs of the

tree-based algorithms, however, do not need normalization. Therefore, we consider both the normalized/non-normalized input ($X$), and compared their cross-validated root mean squared error (RMSE). All the machine-learning models presented in this study were constructed using the scikit-learn library in Python. 6 machine-learning models are trained, including ordinary least squares (LR), K-nearest neighbor regression (KNN), random forest regression (RF), gradient boosting regression with decision trees (GB), neural network (multilayer perceptron) regression (NN), and support vector machine regression (SVR). The LR model minimizes the residual sum of squares between the experimental/observed data points and the predicted data points (MSE). This serves as the benchmark of the other non-linear 5 algorithms. The other 5 algorithms have their hyperparameters optimized using GridSearchCV function based on their MSE after fivefold cross-validation (random 80%:20% training: test split). The cross-validation approximates the testing error, or the error of generalization to related out-of-distribution data. The model hyperparameters were optimized using Grid-SearchCV, and after the models were trained, the root mean squared error (RMSE) was calculated based on fivefold cross-validation. Random forest regression is an ensemble method that works by having multitude of decision trees, where each is constructed independently from a different subsample[52]. Random forest regression resulted in the lowest RMSE, and this fitted model was used to rank the feature importance of the material properties and processing conditions using SHAP (SHapley Additive exPlanations)[35]. The SHAP formula is shown in Eq. (1), where $g$ is the explanation model, $M$ is the maximum coalition size/the number of simplified input features, $\phi_i \epsilon \mathbb{R}$ is the feature attribution for a feature $i$, $z\prime\epsilon\{0,1\}^M$, and $\phi_0$ represents the model output with all the simplified inputs missing.

$$g(z\prime) = \phi_0 + \sum_{i=1}^{M} \phi_i z_i\prime \qquad (1)$$

Our data and trained models are available in the GitHub repository (https://github.com/PV-Lab/capping-layer).

## Data availability
The machine-learning input that supports the findings of this study are available in PubChem 2019 database: https://doi.org/10.1093/nar/gky1033.34 The degradation onset and slope data are available in Supplementary Data 1.

## Code availability
The codes and the data sets used for preprocessing and regression are available in GitHub repository (https://github.com/PV-Lab/capping-layer). The regression algorithms are implemented using scikit-learn[46] and the Shapley values are implemented using SHAP[35] Python library.

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

## Acknowledgements

We thank Kevin Yager and Masafumi Fukuto (Brookhaven National Laboratory) for assistance in synchrotron-based characterization; Rafael Gomez-Bombarelli (MIT), Juan-Pablo Correa-Baena (Georgia Institute of Technology), Ki-Jana B Carter (MIT), and Betar Gallant (MIT), for discussions and input in initial phase of organics screening; REN Zekun Danny and Siyu Isaac Parker TIAN (Singapore-MIT Alliance for Research and Technology) for discussion and input regarding machine-learning methods. This research used 11-BM (CMS) beamline of the National Synchrotron Light Source II, a U.S. Department of Energy (DOE) Office of Science User Facility operated for the DOE Office of Science by Brookhaven National Laboratory under Contract No. DE-SC0012704. Parts of this study were performed at the Harvard University Center for Nanoscale Systems (CNS), a member of the National Nanotechnology Coordinated Infrastructure Network (NNCI), which is supported by the National Science Foundation under NSF award no. 1541959. CNS is part of Harvard University. This work also made use of the MRSEC Shared Experimental Facilities at MIT, supported by the National Science Foundation under award number DMR-1419807. This work was supported by the National Science Foundation (NSF) SusChem Grant CBET-1605547 [N.T.P.H.]; Skoltech Grant 1913/R as part of the Skoltech NGP Program [N.T.P.H.]; TOTAL SA research grant funded through MIT*eI* Sustng Mbr 9/08, RPP [J.T., C.B., Z.L., S.S.]; Alfred Kordelin Foundation and Svenska Tekniska Vetenskaps-akademien i Finland [A.T.]; U.S. Department of Energy (DOE) under Photovoltaic Research and Development (PVRD) program under Award No. DE-EE0007535 [Z.L., F.O.]; the Institute for Soldier Nanotechnology (ISN) Grant W911NF-13-D-0001, the National Aeronautics and Space Administration (NASA) Grant NNX16AM70H [J.J.Y.]; a MISTI-Spain research grant, and Research Mobility Program in the US of UPM [D.F.M.].

## Author contributions

N.T.P.H. and J.T.: fabricated the films. N.T.P.H., J.T., C.B., J.J.Y., R.L., and S.S.: performed in-depth materials characterization. N.T.P.H., A.T., F.O., and C.B.: analyzed the degradation images data and performed machine-learning analysis. N.T.P.H., Z.L., D.F.M., and S.S.: interpreted experimental results. M.G.B., T.B., and S.S.: supervised the study. N.T.P.H., T.B., and S.S.: contributed to the writing the paper.

## Competing interests

The authors declare no competing interests.
