## [Peer Review File · Nature Communications]

Reviewers' comments:

Reviewer #1 (Remarks to the Author):

This manuscript reports a machine learning investigation of layered 3D/2D perovskite structures. The authors have used 21 cations to form a 2D layer on top of 3D perovskite. The degradation mechanism of perovskite layers under accelerated aging conditions of 85% RH, 85°C, correlated with a number of hydrogen-bond donors and the polar surface area of the organic capping layer molecules. They identified top-performing capping layer cation, phenyltriethylammonium iodide, which does not have hydrogen-bonding suggesting capping layer design rules that may further enhance the environmental stability of halide perovskite devices under harsh conditions. The manuscript is in the interests of broad readership; therefore, we recommend for publication after addressing the following queries.

(1). There is no presented framework such as software, and material science design.

(2). The paper has no dataset itself and the network structure with result model weights and the discussion about it. Without this data, the result is nonreproducible.

(3). The authors have an ambitious target to make a model for at least 12 independence parameters, (lines 154-160) but they do not provide any device statistics for the same conditions. As a result, the sustainability of the dataset is doubtful. At the same time, there is no discussion about input data encoding, normalization, etc.

(4). Line 132: It does not matter what type of files the authors use to save images. However, if authors noticed it, it would be nice to get an explanation: why did they use compressed type (.jpg) in some cases and bitmap in others?

(5). Lines 371-373: Is it supposed to be available by the reader/community?

Reviewer #2 (Remarks to the Author):

I found this manuscript to be very interesting and well written. The topic is timely and applies statistical analysis to an important problem in science and engineering. The capping layer results will be of interest to the halide perovskite community and the rapid stability screening approach is of more general interest. I have no hesitation in recommending it to be accepted in Nature Communications. I just have a few minor comments.

a) Figure 1 - I would suggest to make the font sizing more consistent and ensure the schematics will be visible when the figure is scaled down to single column size

b) It is not clear to me what the model is trained to predict, i.e. stability as "yes/no" or the decomposition kinetics (time constant). This could be made clearer in the text

c) I think Figure S1 could be a useful addition to the main text as it conveys the main concept of the study and how striking the colour changes are

d) It would be useful, but not essential, if a repository containing the ML workflow was made available online or provided as SI

Yours sincerely,
Aron Walsh

Reviewer #3 (Remarks to the Author):

To the Authors:

This manuscript describes the use of ML to diagnose the stability of perovskite for photovoltaics. Prior to acceptance of the manuscript, the questions and comments below should be fully addressed:

- 1) According to the manuscript, the authors fabricated 252 distinct types of material data (different material + processing conditions) to training the ML model. But it's not clear how many samples are created and tested for a given material with a certain processing condition to have a significant statistical meaning for the result. This is a critical point of the work, which is currently unclear.
- 2) What exactly is the ML model output? The author claims both the onset time and rate of change are important, but looks like the output of the model is the onset only. Please clarify.
- 3) One of the key steps in ML is feature engineering, and strongly correlated features need to be carefully addressed. Can the author provide any insight or data on how these chemical features of the organic molecules correlate with each other? After all, the community is interested in using ML to learn something about the perovskite material.
- 4) The SEM surface morphology shows the capping layer seals off the grain boundaries on the surface of MAPbI₃ before aging test. Do the authors also have any SEM images showing how the surface looks after aging test so it is more clear how the capping layer helps with the improvement of stability?
- 5) How is the color change of the perovskite defined in a quantitative manner? A detailed description should be provided as photographs are prone to multiple 'artifacts' when comparing different samples.
- 6) The error is very high even for the best model. Thus, one could easily argue that the conclusion presented that PTEAI is "the best" is not corroborated by ML. Please clarify this important point.
- 7) The definition of SHAP use is unclear for nonexperts -- what is difference between high/low values? Keep in mind that most readers will not be ML experts, and actually researchers focused on the development of perovskites.
- 8) How can the Authors assure that the amount of data acquired is sufficient for a valid ML test? The data amount is quite modest...

9) Can the Authors use the model to predict potentially high-performing capping layers, then get experimental data to validate?

10) Figure 3a is unclear/hard to read.

11) A detailed discussion concerning the limitations related to using the hue of the film (red) as stability descriptor for ML is missing.

12) The conclusion contains a claim about suppressing oxygen-containing compound formation in bulk, which is unsupported since XPS is a surface-sensitive technique. Please clarify or add experiments that support this claim.

13) The references below should be added to the manuscript as they refer to very relevant and prior work (in alphabetical order):

- Dordevic et al. *ACS Photonics* **5**, 4888-4895 (2018)

- Howard et al. *Joule* **3**, 325-337 (2019)

- Stoddard et al. *ACS Energy Letters*, online. DOI: 10.1021/acseenergylett.0c00164

Reviewer 1

1. There is no presented framework such as software, and material science design.
2. The paper has no dataset itself and the network structure with result model weights and the discussion about it. Without this data, the result is non-reproducible.
3. At the same time, there is no discussion about input data encoding, normalization, etc.

The reviewer gave an important suggestion to explain the framework of the study, especially the machine learning part and the data pre-processing more thoroughly.

We have added the following text to the method.

“All the features’ values are continuous, which therefore do not need further encoding for the input data. The capping layer material name is not used as one of the features, because its properties have been included in the features instead. Therefore, the encoding done is normalization of the input data for the models. We performed normalization of the model inputs (X) using the the *StandardScaler* algorithm, in the *scikit-learn* library, which calibrates the mean and scales to unit variance. The inputs of the tree-based algorithms, however, do not need normalization. Therefore, we consider both the normalized/ non-normalized input (X), and compared their cross-validated root mean squared error (RMSE).

All the machine learning models presented in this study were constructed using the *scikit-learn* library¹ in Python. 6 machine learning models are trained, including ordinary least squares/ linear regression (LR), K-nearest neighbor regression (KNN), random forest regression (RF), gradient boosting regression with decision trees (GB), neural network (multilayer perceptron) regression (NN), and support vector machine regression (SVR). The LR model minimizes the residual sum of squares between the experimental/ observed data points and the predicted data points (MSE). This serves as the benchmark of the other non-linear 5 algorithms. The other 5 algorithms have their hyperparameters optimized using GridSearchCV function based on their MSE after 5-fold cross-validation (random 80% : 20% training : test split). The cross-validation approximates the testing error, or the error of generalization to related out-of-distribution data.”

Our data and trained models are available in the GitHub repository (<https://github.mit.edu/nhartono/capping-layer>). The parameters for the machine learning algorithms and their RMSE’s are provided below (Table A1 and A2, Figure A1), and included in the supporting information. The LR model presents a significant difference in cross-validated RMSE between the normalized and non-normalized data, while the other algorithms are not affected much by the data pre-processing method.

Regression algorithms	Hyperparameters in scikit-learn	Train RMSE (mins.)	Test RMSE (mins.)	Cross-validated RMSE (mins.)
Linear		61.8	84.0	3.04E13
K-nearest neighbor regression	{'algorithm': 'ball_tree', 'n_neighbors': 3, 'weights': 'uniform'}	48.4	90.3	116.7
Random forest	{'max_depth': 4, 'n_estimators': 30}	46.4	70.8	105.4
Gradient boosting	{'max_depth': 4, 'n_estimators': 40}	39.8	81.9	115.7
Neural network (multilayer perceptron)	{hidden_layer_sizes = (128,256,64), max_iter = 1000, learning_rate = 'constant', solver = 'adam', alpha = 0.01, activation='relu'}	64.3	84.2	161.2
Support vector machine	{'C': 500.0}	41.3	142	97.3

Table A1 (Newly added Table S2 in SI). The machine learning algorithms, their optimized hyperparameters, and their cross-validated RMSE's for *normalized* input (X).

Regression algorithms	Hyperparameters in scikit-learn	Train RMSE (mins.)	Test RMSE (mins.)	Cross-validated RMSE (mins.)
Linear		61.8	84.0	164.3
K-nearest neighbor	{'algorithm': 'ball_tree', 'n_neighbors': 3, 'weights': 'uniform'}	46.4	91.2	148.3
Random forest	{'max_depth': 4, 'n_estimators': 60}	46.5	70.8	104.8
Gradient boosting	{'max_depth': 4, 'n_estimators': 30}	41.7	78.5	112.3
Neural network (multilayer perceptron)	{hidden_layer_sizes =(128,256,64),max_iter=10000, learning_rate='constant', solver='adam', alpha=0.1, activation='relu'}	44.5	89.2	132.4
Support vector machine	{'C': 1000.0}	41.3	142	169.0

Table A2 (Newly added Table S3 in SI). The machine learning algorithms, their optimized hyperparameters, and their cross-validated RMSE's for *non-normalized* input (X).

Figure A1 (Newly added Figure S5 in SI). Cross-validated RMSE for *normalized* input (a) and *non-normalized* input (b). The error bars indicate the standard deviation of the 5-fold cross validation results.

We also run SHAP on the LR, NN, and SVR, for both normalized and non-normalized input data. The complete SHAP results are shown in Appendix I. On the other hand, the SHAP result from RF, GB, and SVR, are similar, showing the reproducibility of the trained ML models. These ML methods are fundamentally different from each other; thus, we interpret this similarity as a strong signal of non-linear variable correlation and importance. The data reproducibility issue, on the other hand, is addressed in the next question.

We have added a new paragraph in the manuscript (line 182-198) to discuss the choice of ML framework, software, input data encoding and normalization for this study. Figures and data supporting the discussion are added to the supporting information.

4. The authors have an ambitious target to make a model for at least 12 independence parameters, (lines 154-160) but they do not provide any device statistics for the same conditions. As a result, the sustainability of the dataset is doubtful.

We agree with the reviewer that providing statistical analysis is key to guarantee reproducibility. In this study we manufactured 260 samples with an average of 11.8 samples per material. We have added the information on statistics of samples into the supporting information (Table A3 and Figure A2, A3), and as discussed below.

Compound	Slope R Mean (mins. ⁻¹)	Slope R Std. Dev. (mins. ⁻¹)	Time-intercept R Mean (mins.)	Time-intercept R Std. Dev. (mins.)	Number of films
BzABr	1.43	0.05	74	51	3
BzAI	1.00	0.35	81	39	12
DABr	1.75	0.72	100	30	12
DAI	1.80	0.43	122	10	12
DMAI	0.90	0.35	91	23	13
FAI					1
GI					1
IDI					1
iPAI					1
tBAI					1
OABr	0.95	0.11	80	46	3
OAI	0.62	0.27	145	45	15
PhABr	1.09	0.43	100	10	3
PhAI	1.39	1.04	52	33	3
TBABr	0.31	0.05	363	60	24
TBAI	0.33	0.08	376	55	28
TPABr	0.44	0.09	374	46	12
TPAI	0.43	0.12	353	66	28
EAI	1.34	0.53	136	49	13
PEAI	0.54	0.19	241	68	12
PTEAI	0.27	0.10	462	115	27
Bare MAPbI ₃	1.29	0.71	107	45	35
Total film samples					260

Table A3 (Newly added Table S4 in SI). The statistics of the film samples for the red (R) slope and red (R) time-intercept. Note that for FAI, GI, IDI, iPAI, tBAI, there is only one film each, fabricated using 10 mM precursor solution concentration and annealed at 100°C.

The films are fabricated using 3 different precursor solutions (5, 10, 15 mM), and after spincoating, they are annealed at 4 different temperatures (50, 75, 100, 125 °C), totaling 12 different processing conditions. The mean and standard deviation of the metrics for each film are shown in **Table A3**.

Table A3 shows a wide range of the mean (52-462 minutes) and standard deviation (10-115 minutes) for various capping materials' time-intercept/ degradation onset. The distribution of these films can also be seen in **Figure A2**. Note that since formamidinium iodide (FAI), guanidinium iodide (GI), imidazolium iodide (IDI), *iso*-propylammonium iodide (iPAI), and *tert*-butylammonium iodide (tBAI) as capping materials have really low time-intercept R, which means they turn yellow readily, and in some cases, faster than the bare MAPbI₃ films, we have fewer data points (only one for specific processing condition: 10 mM precursor solution concentration and 100°C annealing temperature).

Given the relatively large variability, we perform ANCOVA/ analysis of covariance in terms of the red degradation onset (time-intercept), shown in Figure A4. ANCOVA considers the variations in processing conditions, namely the concentration and annealing temperature. At 95% confidence level, which is when the ANCOVA result is less than 0.05, the degradation onset of the most stable capping layer material in this study, PTEAI, is statistically significantly different from other materials and bare MAPbI₃ film. With the same confidence level, the only capping layer materials which have statistically significant difference to the bare MAPbI₃ film are PEAI, TBABr, TBAI, TPABr, and TPAI. Due to large inherent variance in the perovskite materials, the onsets vary greatly, and it is harder to get statistically significant results. Note that we have only made conclusions for statistically significant results.

Figure A2 (Newly added Figure S34 in SI). The time-intercept R data/ red onset/ *xinterp_r* (a) and the slope R data/ *param1_r* (b) for the various capping layer materials. Each capping material's boxplot represents the data points across all the processing conditions with various precursor solution concentrations and annealing temperatures.

Figure A3 (Newly added Figure S8 in SI). The ANCOVA statistical analysis for time-intercept R data/ red onset/ *xinterp_r* at 95% confidence level. The sigma values lower than 0.05 show statistically significant differences (purple color) The most stable capping layer, PTEAI, is statistically significantly different in comparison to other materials and bare MAPbI₃.

The goal of the study is building the ML model as a robust prescriptive model for interpreting the degradation test result, instead of a fully generalizable predictive model, which requires a deeper molecular featurization. The dataset size, the number of features, the number of models being tested, and the cross-validation in this study are sufficiently robust. The good cross-validated performance, between the different models, given the inherent variability of perovskites, along with the similar SHAP results, gives confidence in the model interpretation. A similar prescriptive approach has been applied successfully to other fields with similar dataset size, or even highly-qualitative or biased datasets.^{2,3}

- Line 132: It does not matter what type of files the authors use to save images. However, if authors noticed it, it would be nice to get an explanation: why did they use compressed type (.jpg) in some cases and bitmap in others?

We appreciate the reviewer for asking this question and have updated the manuscript in line 412-417 to explain this point.

JPEG is a compressed file format potentially losing some color information, while BMP is a non-compressed format but a lot larger in size. All the image analysis data in this study is done using JPEG images as it allows a faster image processing than using the bitmap format. However, we also confirmed that the JPEG images, and their quality is as good as their bitmap images for the purpose of this study, with negligible average difference (< 1) in RGB channel values (Figure A4). Therefore, we decided to use JPEG to process the degradation test data.

Figure A4 (Figure S28 in SI). The average difference of respective RGB color values between JPEG and BMP formats of the 28 random samples' aging test images, showing negligible difference (< 1).

- Lines 371-373: Is it (the accelerated aging test data) supposed to be available to the readers/ community?

Yes, the accelerated aging test result is available in .csv format, and has been added to the supporting information. The ML workflow is available in the Jupyter notebook (.ipynb) format. Here is the address for the online repository where the Jupyter notebook can be found: <https://github.mit.edu/nhartono/capping-layer>.

Reviewer 2

1. Figure 1 - I would suggest to make the font sizing more consistent and ensure the schematics be visible when the figure is scaled down to single column size.

We thank the reviewer for pointing this out. We updated Figure 1 in the manuscript, and ensured more consistent font sizing.

2. What is the model trying to predict (e.g. stability as a “yes/no” or the decomposition kinetics (time constant), could be made clearer in the text)?

The stability descriptor is extracted the following way.

- a. We started by performing accelerated degradation test until all the materials decompose, instead of set a fixed period of time and examine whether a decomposition has taken place. Hence, all the materials would have been classified as “yes” in a conventional binary classification model.
- b. We built a model to predict the degradation onset, *i.e.* the time when the decomposition starts. This is a prediction on how long will a film remain structurally stable (in black perovskite phase) under 85% relative humidity/ 85°C test condition. Considering that the color change corresponds to the change of crystal structure from MAPbI_3 into PbI_2 ,^{4,5} as soon as the yellow spots appear, the films have started to degrade. The variable, time, is a continuous variable.
- c. We can also predict kinetics of the decomposition (time constant) by extracting the rate of change in color after the degradation starts (“red slope” in Figure A5), however, in this study, we focused on only taking the time at the onset when the degradation starts as our output.

Figure A5 (Newly added Figure S3 in SI). The red onset (the time-intercept of the degradation curve) and the red slope (the slope of the sharpest change in the degradation curve) for the red color.

We have clarified this point in the manuscript in line 143-150, and in the supporting information.

3. I think Figure S1 could be a useful addition to the main text as it conveys the main concept of the study and how striking the colour changes are.

The color change progress is added into Figure 1 in the main text. We also show three films (bare MAPbI₃, OABr-capped, and PTEAI-capped) at 2 different time points in Figure 2d of the main text.

4. A repository containing the ML workflow, was made available online/ provided as supporting information.

Yes, the accelerated aging test result is available in .csv format, and has been added to the supporting information. The ML workflow is available in the Jupyter notebook (.ipynb) format. Here is the address for the online depository where the Jupyter notebook can be found: <https://github.mit.edu/nhartono/capping-layer>.

Reviewer 3

1. According to the manuscript, the authors fabricated 252 distinct types of material data (different material + processing conditions) to training the ML model. But it's not clear how many samples are created and tested for a given material with a certain processing condition to have a significant statistical meaning for the result. This is a critical point of the work, which is currently unclear.

We thank the reviewer for bringing this important point to our attention. In this study we have manufactured 260 samples, with 21 types of materials (22 including the bare MAPbI₃ film) data. To ensure the robustness, reproducibility and sustainability of our dataset and model, we performed a detailed statistical analysis, which is discussed in the page 3-6 above (Reviewer 1, question 4). We have added the results of the statistical analysis in supporting information, Figure S7 and Table S5.

2. What exactly is the ML model output? The author claims both the onset time and rate of change are important, but looks like the output of the model is the onset only. Please clarify.

We thank the reviewer for bringing this point to our attention, and we have clarified this important point in the manuscript (line 125-130, 143-150 of the main text and supporting information).

The stability descriptor is extracted the following way.

- a. We started by performing accelerated degradation test until all the materials decompose, instead of set a fixed period of time and examine whether a decomposition has taken place. Hence, all the materials would have been classified as “yes” in a conventional binary classification model.
- b. We built a model to predict the degradation onset, *i.e.* the time when the decomposition starts. This is a prediction on how long will a film remain structurally stable (in black perovskite phase) under 85% relative humidity/ 85°C test condition. Considering that the color change corresponds to the change of crystal structure from MAPbI₃ into PbI₂,^{4,5} as soon as the yellow spots appear, the films have started to degrade. The variable, time, is a continuous variable.
- c. We can also predict kinetics of the decomposition (time constant) by extracting the rate of change in color after the degradation starts (“red slope” in Figure A5), however, in this study, we focused on only taking the time at the onset when the degradation starts as our output.

After the degradation onset data is compiled, we can combine it with the materials data from PubChem 2019 database. Because the data is continuous (instead of “binary”, “yes”/“no”), we can use the regression models. The regression models work by fitting the input (X) consisting of

molecular properties of the capping layer materials and their processing conditions (the precursor solution concentration and the annealing temperature), into the output (y), consisting of the degradation parameters (in this study, the time-intercept of red color value/ R).

In this study, we choose the time-intercept of the degradation curve as the sign of degradation starting point in the film (the “red onset”).

Figure A6 shows that there is a weak inverse correlation between the red (R) slope with the red (R) time-intercept (Pearson correlation value = -0.61). The more stable capping materials with high time-intercepts (above 260 minutes) show a stronger inverse correlation than the less stable materials (time-intercepts below 250 minutes). For more stable materials, we can also conclude that they tend to have lower slope/ slower ‘yellowing’ speed.

Figure A6 (Newly added Figure S35 in SI). The correlation between the time-intercept R the slope of R, with the Pearson correlation value of -0.61.

Figure A8 shows that the change in blue color is insignificant in comparison to the red and green color change. This is supported by the fact that the MAPbI₃ films undergo ‘yellowing’ when they turn into PbI₂,⁶ which is also shown on the x-ray diffraction (XRD) results, shown in Figure 4b in the main text. Yellow comes from red and green light. Therefore, we can choose between red and green to describe the degradation of the films. Figure A9 shows how the red and green color correlates for both the time-intercept and the slope of degradation curves. Pearson correlation values for time-intercept and slope are 0.99 and 0.96, respectively, indicating that we can choose either color as our dataset’s degradation descriptor. Hence, in this study, we choose to describe the degradation process using the time-intercept of the red color curve.

Figure A7 (Newly added Figure S37 in SI). A typical red, green, blue (RGB) curve from bare MAPbI₃ and PTEAI-capped films.

Figure A8 (Newly added Figure S4 in SI). The correlation between the time-intercept R and G (a), with Pearson correlation value of 0.99 and between the slope of R and G (b), with Pearson correlation value of 0.96.

3. One of the key steps in ML is feature engineering, and strongly correlated features need to be carefully addressed. Can the author provide any insight or data on how these chemical features of the organic molecules correlate with each other? After all, the community is interested in using ML to learn something about the perovskite material.

The reviewer raises an important concern regarding feature correlation, that had not been discussed sufficiently in the manuscript. This discussion has been added to the supporting information.

We only include 14 descriptors/ features/ molecular properties from the PubChem 2019 database. There are more molecular features from the database that are not included, because the molecules that we explore are highly correlated in those features, such as the number of Hydrogen bond acceptors; the number of other atoms (O, Cl, F); formal charge; atom, bond stereocenter count; and covalently-bonded unit count.

The molecular features' correlations between each other are shown in Figure A9, using Pearson correlation value, which ranges between -1.00 and 1.00. There are few things that should be noted:

1. The Pearson correlation value between the number of Br and I atoms is -1.00, and this is due to the fact that the X-site anion is either Br or I.
2. The molecular weight is correlated strongly (> 0.9) with the number of heavy atoms, and the number of C atoms. The number of C atoms also correlates with the complexity, partition coefficient ($x \log P$), the number of rotatable bonds, and the number of Hydrogen atoms. These molecular properties are all related to each other.
3. The topological polar surface area and the number of Hydrogen bond donor, which come as the most important feature, correlate with Pearson correlation value of 0.81.

SHAP is more robust to correlated features⁷ than traditional interpretability methods, due to the game-theoretic nature of feature attribution. However, if strong multicollinearity exists (*i.e.* very high shared variance among features causing significant variations in model training), highly correlated features might be incorrectly attributed feature importance, according to how the model is fitting the data. To diminish these spurious attributions, we train various different models *via* cross-validation and make our final decision based on contrasting the feature ranking (see Appendix I). The consistency among models, along with testing using cross validation, allows us to physically interpret the models. The most robust approach, which is incompatible with our exploratory goals, could have been to perform causal inference using randomized control trials or experiments in a more limited compound space. The tradeoff, of course, is the slow and poor exploration of the material space.

Based on the SHAP feature importance rank (Figure A10) for the random forest regressor, to pick the organic A-site cation, the most important capping molecular features are the following.

1. The first thing that we should look at is how polar the area of the cation. The less polar the cation is, the more stable it is as a capping layer material for MAPbI₃ film. Polarity has relatively low correlation with other variables, the highest correlation is with the number of hydrogen-bond donor (Pearson correlative coefficient is 0.81).
2. Molecular weight is the next important feature, which means the cation size matter. The higher the molecular weight is, and the more complex (branches, different subgroups) the organic cation is, the more the capped-films' stability improves. The molecular weight has the highest correlations with the number of carbon atoms (Pearson correlative coefficient is 0.95), the number of heavy atoms (0.95), complexity (0.89), the number of hydrogen atoms (0.89), and partition coefficients (0.87).

3. The capping material precursor solution concentration is the third most important feature, where higher precursor solution concentration improves stability of the capped films. As the X-ray diffraction (XRD) data has shown, the capping materials form low dimensional perovskite, by reacting with the excess PbI_2 coming from the MAPbI_3 film.⁸ Therefore, the higher the solution concentration is, the thicker the capping layer forms, and the more resistant the film is from the environmental stress. However, a thicker capping layer might lead to the reduction in device performance,⁹ which warrants a future study considering the trade-off between stability and performance of capped devices.

Figure A9 (Newly added Figure S10 in SI). The Pearson correlation coefficients for the machine learning input (X), which includes processing conditions and the molecular properties. The negative values imply an inverse relationship.

Figure A10 (Updated Figure 3d in main text). The feature importance rank generated using SHAP and random forest regression (RF, *non-normalized*) for the molecular properties determining the time-intercept/ degradation onset of the capped films. The *x*-axis corresponds to the model output (higher means improving the stability, and vice versa).

4. The SEM surface morphology shows the capping layer seals off the grain boundaries on the surface of MAPbI₃ before aging test. Do the authors also have any SEM images showing how the surface looks after aging test so it is more clear how the capping layer helps with the improvement of stability?

Thank you for bringing up the question about SEM. We have included the SEM surface morphology pre- and post-degradation in Figure A11 (and shown in Figure S14 of the supporting information). Looking at the pre-degraded images (at 0 minute), the grain boundaries of the capped films are ‘blanketed’ with another material. The 460-minute- and 640-minute-degraded films, however, look similar between the bare and capped films.

Figure A11 (Figure S22 in SI). Aging tests, microstructure: SEM images for bare, TPAI (tetrapropylammonium iodide)-capped, and PTEAI (phenyltriethylammonium iodide)-capped MAPbI₃ films at 0, 460, and 640 minutes of degradation. Scale bar: 200 nm.

5. How is the color change of the perovskite defined in a quantitative manner? A detailed description should be provided as photographs are prone to multiple ‘artifacts’ when comparing different samples.

The reviewer expressed a relevant concern since the observed color depends on the observer and illumination in addition to the color of the specimen. We have paid attention to design our procedure so that it ensures the tracked colors are repeatable (the results can be repeated in our laboratory at another time) and reproducible (the results can be obtained in another laboratory).

Quantitative color analysis is utilized in other fields, e.g. in food industry quality control, and therefore we could utilize procedures in these fields while designing our method.

We quantified the colors by keeping the illumination conditions in our setup as constant as possible and by using color calibration to transform the colors to a stable reference color space. We have described the whole procedure in detail for your information in Appendix II.

6. The error is very high even for the best model. Thus, one could easily argue that the conclusion presented that PTEAI is "the best" is not corroborated by ML. Please clarify this important point.

We have added a paragraph in the main text (line 196-201, 206-224) to clarify this important point. The conclusion that PTEAI gives the best performance is corroborated by ML, for the following reasons:

- a. Error analysis: Perovskites are known to have high variability in stability,^{10,11} the performance of bare MAPbI₃ (degradation onset standard deviation is 45 minutes, or 42% of the onset mean) confirmed this. Even after applying capping layers, the variability is still high. Hence, the comparison between the capping layer materials' degradation results are important.
- b. Statistical analysis: We repeat the tests multiple times with total of 260 samples. (see page 1-3, Reviewer 1, question 1-3 and Figure A3 for the detailed statistical analysis). We perform ANCOVA/ analysis of covariance in terms of the red degradation onset (time-intercept) (Figure A3). ANCOVA considers the variations in processing conditions, namely the concentration and annealing temperature. At 95% confidence level, which is when the ANCOVA result is less than 0.05, the degradation onset of the most stable capping layer material in this study, PTEAI, is statistically significant in comparison to other materials and bare MAPbI₃ film. With the same confidence level, the only capping layer materials which are statistically significant in comparison to the bare MAPbI₃ film are PEAI, TBABr, TBAI, TPABr, and TPAI.
- c. Feature analysis: The SHAP result shows that the low number of Hydrogen bond donor and small topological polar surface area are the most important features leading to a more stable film. PTEAI indeed has lower number of Hydrogen bond donor (0) and smallest polar surface area (0 Å²) among the materials we explore. These two features also appear as the top important features on other trained ML models (see Appendix I), confirming the robust results.

We completely removed the best performing capping layer material, PTEAI from the capping layer candidates, and trained the models with the remaining 20 capping layer candidates. The feature importance rank result is shown in Figure A11, showing that the number of Hydrogen-bond donor and the topological polar surface area come out as the two most important features in random forest regression model. The model still predicts materials with low hydrogen-bond donor and polar surface area to be as the top most stable capping layer material. This nevertheless points us to PTEAI as a promising candidate, since PTEAI has hydrogen bond donor of 0, and topological polar surface of 0 Å². Based on this model, the predicted degradation onset for PTEAI is 355.9 minutes, the mean experimental degradation onset for PTEAI in our study is 462 minutes, with the standard deviation of 115.

7. The definition of SHAP use is unclear for nonexperts -- what is difference between high/low values? Keep in mind that most readers will not be ML experts, and actually researchers focused on the development of perovskites.

The explanation of SHAP is updated in the main text of the manuscript (line 202-212) for clarification. SHAP is used to help understanding how low/ high value of features impacts the stability. The model input (X) which is also called features, consists of processing conditions and molecular properties. Figure A10 shows the SHAP result from random forest regression model. The dark color represents low feature value, and the light color represents high feature value. The x -axis represents the SHAP value (impact on the model output), ranging from negative to positive values. If the feature values fall in the positive SHAP value, that particular feature values improve the stability, and vice versa. The low feature value (dark color dots) for the number of Hydrogen bond donors and the topological polar surface area, for instance, improves the SHAP value, which is synonymous with higher stability.

8. How can the Authors assure that the amount of data acquired is sufficient for a valid ML test? The data amount is quite modest.

Sufficiency of the data is indeed important for getting meaningful machine learning results, and we acknowledge that there are many existing machine learning methods that require thousands of data points and are therefore out of reach in many experimental science fields. To ensure our dataset of 260 data points (new data has been added, which is updated in line 178-179 of the manuscript, and this number is larger than most experimental datasets presented in perovskite solar cell field) produces meaningful results, we tested SHAP with various models, and considered as relevant only those findings that could be repeated with any of them, namely: gradient boosting regression with decision trees, random forest regression, multi-layer perceptron, and support vector machine regression. The SHAP result for other models are shown in Appendix I. We also try to exclude some data points (PTEAI-capped data), and both algorithms also show the same top 4 SHAP ranks (Table A4).

The goal of the study is building the ML model as a robust prescriptive model for interpreting the degradation test result, instead of a fully generalizable predictive model, which requires a deeper molecular featurization. The dataset size, the number of features, the number of models being tested, and the cross-validation in this study are sufficiently robust. The good cross-validated performance, given the inherent variability of perovskites, along with the similar SHAP results, gives confidence in the model interpretation. A similar prescriptive approach has been applied successfully to other fields with similar dataset size, or even highly-qualitative or biased datasets.^{2,3}

Machine learning models	Top 5 SHAP features, dataset: all	Top 5 SHAP features, dataset: excluding PTEAI-capped
Random forest regression	 1. Top. polar surface area 2. # H-bond donor 3. Molecular weight 4. Concentration 5. Partition coefficient 	 1. # H-bond donor 2. Top. polar surface area 3. Molecular weight 4. Concentration 5. Annealing temperature
Gradient boosting regression with decision trees	 1. Top. polar surface area 2. # H-bond donor 3. Molecular weight 4. Concentration 5. Annealing temperature 	 1. # H-bond donor 2. Top. polar surface area 3. Molecular weight 4. Annealing temperature 5. Complexity
Linear regression (non-normalized)	 1. Molecular weight 2. # Heavy atom 3. # H atom 4. # Complexity 5. # I atom 	 1. Molecular weight 2. # H atom 3. # Heavy atom 4. Complexity 5. # Rotatable bond

Table A4 (Newly added Table S6 in SI). The top 5 SHAP features for linear regression, random forest regression, and gradient boosting regression with decision trees, with the complete dataset and PTEAI-capped-excluded dataset.

9. Can the Authors use the model to predict potentially high-performing capping layers, then get experimental data to validate?

We have performed a test following the same workflow for predicting potentially high-performing capping layer material and examine its experimental performance as suggested by the reviewer:

We completely removed the best performing capping layer material, PTEAI from the capping layer candidates, and trained the models with the remaining 20 capping layer candidates. The feature importance rank result is shown in **Figure A12**, showing that the number of Hydrogen-bond donor and the topological polar surface area come out as the two most important features in random forest regression model. The model still predicts materials with low hydrogen-bond donor and polar surface area to be as the top most stable capping layer material. This nevertheless points us to PTEAI as a promising candidate, since PTEAI has hydrogen bond donor of 0, and topological polar surface of 0 Å². Based on this model, the predicted degradation onset for PTEAI is 355.9 minutes, the mean experimental degradation onset for PTEAI in our study is 462 minutes, with the standard deviation of 115.

We also used the model developed in this study to predict literature reports caffeine (264.2 minutes), theobromine (121.5 minutes), and theophylline (103.2 minutes) (**Figure A12**).

Figure A11 (Newly added Figure S16 in SI). The feature importance rank generated using SHAP and random forest regression (RF, *non-normalized*) for the molecular properties determining the time-intercept/ degradation onset of the capped films, and excluding the top stable capping layer material, PTEAI. The x -axis corresponds to the model output (higher means improving the stability, and vice versa).

Figure A12 (Newly added Figure S36 in SI). The time-intercept R data/ *xinterp_r* for the trained data, excluding PTEAI (a) and the prediction based on the trained random forest regressor (b). Each capping material's boxplot represents the data points across all the processing conditions with various precursor solution concentrations and annealing temperatures.

The goal of this study is to identify features of capping layer materials, which have the most significant effects on the stability of perovskite thin films. To construct our model, we synthesized and experimentally tested capping layer materials with all the organic precursors available to us. Further model interpolation will require a lot more data with new organic molecules which may be unavailable commercially, a more advanced featurization, and a computational screening. These are beyond the scope of this study. However, to start looking for a new capping layer material, we predict and hope our study can inspire organic chemists to synthesize a cation with quaternary ammonium group and non-polar surface area, because the top stable capping layer materials, such as PTEAI, TBAI, and TPAI, have such ammonium group. Their solubility in the solvent (isopropyl alcohol/ IPA) also needs to be considered, to ensure that they can be dissolved into at least 1 mM precursor solution.

10. Figure 3a is unclear/hard to read.

We have updated **Figure 3a** in the main text so that it is clearer.

11. A detailed discussion concerning the limitations related to using the hue of the film (red) as stability descriptor for ML is missing.

We thank the reviewer for bringing this point to our attention. We have addressed the choice of using red color in the reply to question 2 from Reviewer 3 in page 10-12 above. The fact that the perovskite degrades into yellow-phase PbI_2 is the sole reason why choosing either red or green color to characterize the film degradation works. If we are working with different types of perovskite materials (Bismuth- or Antimony-based, for instance), the degradation product will no longer be the same, and hence, the red color usage to characterize the degradation will no longer be valid.

Another limitation comes from the fact that we are using MAPbI_3 as the perovskite absorber layer, which has a distinct shape, similar to a step function, as shown in Figure A5 and Figure A7. This allows us to characterize degradation by picking the degradation onset and slope. If we use another material as the absorber, for instance the multi-cation perovskites, it is likely that we do not get the same shape, and thus, we need to find another way to characterize how much the samples have degraded over time. We have added this paragraph in supporting information.

12. The conclusion contains a claim about suppressing oxygen-containing compound formation in **bulk**, which is unsupported since XPS is a surface-sensitive technique. Please clarify or add experiments that support this claim.

Thank you for the reviewer's comments regarding the fact that XPS is a surface-sensitive technique, and the adjustment has been made in the manuscript, between line 325-332, and line 365-368. That implies, the XPS result corresponds to the formation of oxygen-containing compound in capping layer, instead of in bulk.¹²

13. The references below should be added to the manuscript as they refer to very relevant and prior work (in alphabetical order):

- 1) Dordevic et al. *ACS Photonics* **5**, 4888-4895 (2018)
- 2) Howard et al. *Joule* **3**, 325-337 (2019)
- 3) Stoddard et al. *ACS Energy Letters*, online. DOI: 10.1021/acsenergylett.0c00164

Thank you for pointing out relevant works, they have been added in the references section.

Reviewer 4

1. Use of color as a proxy for measuring onset of degradation. There is little to no justification provided for this proxy choice. Furthermore, I imagine that the capping layers could themselves influence the colour of the samples, obscuring or enhancing the signal associated with degradation. Given the already statistically rather noisy estimates and fits presented in Figure 3 I think that this is a real concern.

The reviewer expressed a relevant concern regarding the color as a proxy for measuring onset of degradation. The perovskite field has previously used color as a proxy for a film's degradation.^{13–}

¹⁶

The capping layer did influence the starting color of the samples, as shown in Figure A13, however, they all started with black perovskite phase and turned into yellow phase of PbI_2 when they degraded. The extracted red, green (RG) values from the image also show different starting points (Figure A14). Over time, the RG values increase, as the film degrades and changes its color to yellow, which are the same across all films with different capping layer materials. Additionally, various materials' RG curve shape is similar, which resembles a step function. This study focuses on the structural change from black perovskite phase into yellow PbI_2 phase, and we only need the camera instrument to be able to capture the change in color.

However, it should be noted that if we are working with different types of perovskite materials (Bismuth- or Antimony-based, for instance), the starting degradation points will be totally different. The degradation product will no longer be the same, and hence, the red RG values usage to characterize the degradation will no longer be valid.

Figure A13 (Figure S1 in SI). Raw, compiled results from accelerated aging tests. Comparison between average RGB values from degradation of bare MAPbI₃, with capping layers fabricated at optimum condition that gives maximum onsets: formamidinium iodide (FAI), guanidinium iodide (GI), ethylammonium iodide (EAI), dimethylammonium iodide (DMAI), *iso*-propylammonium iodide (iPAI), imidazolium iodide (IDI), *tert*-butylammonium iodide (tBAI), phenylammonium iodide (PhAI), phenylammonium bromide (PhABr), benzylammonium iodide (BzAI), benzylammonium bromide (BzABr), phenylethylammonium iodide (PEAI), *n*-octylammonium iodide (OAI), *n*-octylammonium bromide (OABr), phenyltriethylammonium iodide (PTEAI), dodecylammonium iodide (DAI), dodecylammonium bromide (DABr), tetrapropylammonium iodide (TPAI), tetrapropylammonium bromide (TPABr), tetrabutylammonium iodide (TBAI), and tetrabutylammonium bromide (TBABr). Depending on the processing conditions, the yellowing onset of the films happened ± 30 minutes.

Figure A14 (Figure 2c in main text). The time-dependent red and green values of camera images, for bare MAPbI₃ control material, OABr, used in state-of-the-art high-efficiency devices, and PTEAI, our best-performing material in this study.

The degradation from black perovskite phase to yellow PbI₂ phase is also confirmed by x-ray diffraction results (Figure A14).

Figure A15 (Figure S21 in SI). Aging tests, changes of phase: Powder x-ray diffraction (XRD) data for bare MAPbI₃, TPAI-capped, and PTEAI-capped MAPbI₃. The amount of PbI₂ in the fresh samples of capping layers is reduced in comparison to fresh bare MAPbI₃.

We also did a procedure to calibrate the color on the images, as shown in Appendix II.

The statistical analysis of the samples has been addressed in question 4 from Reviewer 1, in page 3-6 above. The noisy fitting results are caused by the variability in 12 different synthesis conditions, in addition to inherent high variability in the bare MAPbI₃'s degradation profile^{10,11} (standard deviation of red onset across 35 bare MAPbI₃ samples \approx 45 minutes). The high error and noisy data have also been addressed in question 6 of Reviewer 3 in page 17 above. Even though

the noise is high, the degradation onset results from the most stable capping layer, PTEAI, are statistically significant with 95% confidence level (ANCOVA) in comparison to other materials, including the bare MAPbI₃ film.

The figures above have been added to the supporting information.

2. The study does not use the ML procedure to design any new materials. The capping layer selected for further study PTEAI is in fact the best performing capping layer in the training set used to fit the ML models. While it does fit the criteria suggested by the SHAP analysis there is no sense that the ML was used to guide design here. Maybe the authors could use their criteria to rapidly screen databases and propose some other champion materials, but with zero h-bond acceptors and PSA of 0 it kind of suggests that PTEAI might not be possible to beat.

We thank the reviewer for bringing this important point to our attention. The goal of the study is building the ML model as a robust prescriptive model for interpreting the degradation test result and to create, instead of a fully generalizable predictive model, which requires a deeper molecular featurization. The dataset size, the number of features, the number of models being tested, and the cross-validation in this study are sufficiently robust. The good cross-validated performance, given the inherent variability of perovskites, along with the similar SHAP results, gives confidence in the model interpretation. A similar prescriptive approach has been applied successfully to other fields with similar dataset size, or even highly-qualitative or biased datasets.^{2,3}

The study uses ML procedure to model the relationships between the molecular properties of A-site cations in the capping layer materials, with their aging test result. SHAP uses the trained models to ‘rank’ the molecular properties, giving us ideas about which properties affect the stability the most. This is important in the perovskite solar cells field, because a lot of LD perovskites work as capping layer, but we do not know which one gives the *best* stability. This study aims to give the guidance on what kind of A-site cations we need to use in capping layer.

Based on the feature importance rank (Figure A10) coming from the random forest regressor and SHAP, to pick the organic A-site cation, the most important capping molecular features are the following.

1. The first thing that we should look at is how polar the area of the cation. The less polar the cation is, the more stable it is as a capping layer material for MAPbI₃ film.
2. Molecular weight is the next important feature, which means the cation size matter. The higher the molecular weight is, and the more complex (branches, different subgroups) the organic cation is, the more the capped-films’ stability improves.
3. The capping material precursor solution concentration is the third most important feature, where higher precursor solution concentration improves stability of the capped films. As the X-ray diffraction (XRD) data has shown, the capping materials form low dimensional perovskite, by reacting with the excess PbI₂ coming from the MAPbI₃ film.⁸ Therefore, the

higher the solution concentration is, the thicker the capping layer forms, and the more resistant the film is from the environmental stress. However, a thicker capping layer might lead to the reduction in device performance,⁹ which warrants a future study considering the trade-off between stability and performance of capped devices.

Based on this guideline, some capping layer materials we could explore are bigger molecules with quaternary ammonium group (NR_4^+), where R is an alkyl or an aryl group, for instance, N,N,N-trimethylnaphthalen-1-aminium iodide (CAS number: 2350-79-0, Figure A16).

Figure A16 (Newly added Figure S11 in SI). N,N,N-trimethylnaphthalen-1-aminium iodide, as a possible capping layer material to explore.

We also have tried excluding PTEAI in the dataset for training the ML models, and the SHAP results are similar with the one including PTEAI. We have addressed this in question 9 from Reviewer 3 in page 19-21 above.

3. What is the difference between TPAI and PTEAI - I mean in terms of the descriptors suggested as important. I had no concept of why these two capping layers were chosen for comparison.

Thank you for asking to clarify this important point. We have analyzed the correlation of the descriptors/ features (Figure A9) by using Pearson correlation value, which ranges between -1.00 and 1.00. There are few things that should be noted:

1. The Pearson correlation value between the number of Br and I atoms is -1.00, and this is due to the fact that the X-site anion is either Br or I.
2. The molecular weight is correlated strongly (> 0.9) with the number of heavy atoms, and the number of C atoms. The number of C atoms also correlates with the complexity, partition coefficient ($x \log P$), the number of rotatable bonds, and the number of Hydrogen atoms. These molecular properties are all related to each other.
3. The topological polar surface area and the number of Hydrogen bond donor, which come as the most important feature, correlate with Pearson correlation value of 0.81.

Tetrapropylammonium iodide (TPAI, $\text{C}_{12}\text{H}_{28}\text{NI}$) has the same number of carbon, nitrogen, and iodine as phenyltriethylammonium iodide (PTEAI, $\text{C}_{12}\text{H}_{28}\text{NI}$), and thus, they have similar molecular weight. However, they have very different structure, complexity level, the number of hydrogen bonding donor, and polarity. By comparing these two capping layer materials with very

close molecular weight values but different structure, we would like to probe if the stability outcome is similar or different between similar-size molecules, and hence, we compare the two side-by-side in various characterizations in this study. This explanation has been added in line 93, 261-262.

4. Paragraph on lines 234 to 238. I have some problems with this analysis. First I can't see that fig 4c shows the capping layer sealing off grain boundaries - I can simply see a capping layer probably coating the entire surface. Second sure the grains change after degradation, but I find it hard to state any qualitative or quantitative difference between the samples with or without the capping layers, much less do I see any necessary connection to the capping layers sealing off the grain boundaries. I don't think that this analysis is particularly meaningful.

The reviewer raises an important point regarding the SEM results of the capped film versus the bare MAPbI₃ film. The capping layer indeed coats the entire surface, including grain boundaries.

We have modified the manuscript (line 27, 285-286) as the following.

“Comparing the surface morphology of MAPbI₃ and PTEAI-capped MAPbI₃, we noticed a difference at pre-degradation time point, where the capping layers “coat” the surface of MAPbI₃, including the grain boundaries, as shown in Figure 4c. As degradation occurs, and the surface reacts with the high-humidity environment at elevated temperature, the grains change and more pinholes appear, as shown in supporting information Figure S23.”

References

1. Pedregosa, F. *et al.* Scikit-learn: Machine Learning in Python. (2012).
2. Jia, X. *et al.* Anthropogenic biases in chemical reaction data hinder exploratory inorganic synthesis. *Nature* vol. 573 251–255 (2019).
3. Chen, N. C., Drouhard, M., Kocielnik, R., Suh, J. & Aragon, C. R. Using machine learning to support qualitative coding in social science: Shifting the focus to ambiguity. *ACM Trans. Interact. Intell. Syst.* **8**, 1–20 (2018).
4. Aristidou, N. *et al.* Fast oxygen diffusion and iodide defects mediate oxygen-induced degradation of perovskite solar cells. *Nat. Commun.* **8**, 1–10 (2017).
5. Fan, Z. *et al.* Layer-by-Layer Degradation of Methylammonium Lead Tri-iodide Perovskite Microplates. *Joule* **1**, 548–562 (2017).
6. Wang, S., Jiang, Y., Juarez-Perez, E. J., Ono, L. K. & Qi, Y. Accelerated degradation of methylammonium lead iodide perovskites induced by exposure to iodine vapour. *Nat. Energy* **2**, 1–8 (2017).
7. Lundberg, S. M. & Lee, S.-I. A Unified Approach to Interpreting Model Predictions. 4765–4774 (2017).
8. Gao, P., Bin Mohd Yusoff, A. R. & Nazeeruddin, M. K. Dimensionality engineering of hybrid halide perovskite light absorbers. *Nat. Commun.* **9**, 5028 (2018).
9. Gao, L. *et al.* Improved Environmental Stability and Solar Cell Efficiency of (MA,FA)PbI₃ Perovskite Using a Wide-Band-Gap 1D Thiazolium Lead Iodide Capping Layer Strategy. *ACS Energy Lett.* 1763–1769 (2019) doi:10.1021/acsenergylett.9b00930.
10. Tiihonen, A. *et al.* Critical analysis on the quality of stability studies of perovskite and dye solar cells. *Energy Environ. Sci.* **11**, 730–738 (2018).
11. Wang, R. *et al.* A Review of Perovskites Solar Cell Stability. *Adv. Funct. Mater.* **29**, 1808843 (2019).
12. Sun, Q. *et al.* Role of Microstructure in Oxygen Induced Photodegradation of Methylammonium Lead Triiodide Perovskite Films. *Adv. Energy Mater.* **7**, 1700977 (2017).
13. Stoddard, R. J. *et al.* Forecasting the Decay of Hybrid Perovskite Performance using Optical Transmittance or Reflected Dark Field Imaging. *ACS Energy Lett.* acsenergylett.0c00164 (2020) doi:10.1021/acsenergylett.0c00164.
14. Spanopoulos, I. *et al.* Uniaxial Expansion of the 2D Ruddlesden–Popper Perovskite Family for Improved Environmental Stability. *J. Am. Chem. Soc.* **141**, 5518–5534 (2019).
15. Hashmi, S. G. *et al.* Long term stability of air processed inkjet infiltrated carbon-based printed perovskite solar cells under intense ultra-violet light soaking. *J. Mater. Chem. A* **5**, 4797–4802 (2017).
16. Hashmi, S. G. *et al.* Air Processed Inkjet Infiltrated Carbon Based Printed Perovskite Solar Cells with High Stability and Reproducibility. *Adv. Mater. Technol.* **2**, 1600183 (2017).
17. Menesatti, P. *et al.* RGB Color Calibration for Quantitative Image Analysis: The “3D Thin-Plate Spline” Warping Approach. *Sensors* **12**, 7063–7079 (2012).

Appendix I

Figure A17 (Newly added Figure S6 in SI). The feature importance rank generated using SHAP and linear-regression (LR, *non-normalized*) for the molecular properties determining the time-intercept/ degradation onset of the capped films. The *x*-axis corresponds to the model output (higher means improving the stability, and vice versa).

Figure A18 (Newly added Figure S12 in SI). The feature importance rank generated using SHAP and multi-layer perceptron neural network (NN, *non-normalized*) for the molecular properties determining the time-intercept/ degradation onset of the capped films. The *x*-axis corresponds to the model output (higher means improving the stability, and vice versa).

Figure A19 (Newly added Figure S13 in SI). The feature importance rank generated using SHAP and support vector machine regression (SVR, *non-normalized*) for the molecular properties determining the time-intercept/ degradation onset of the capped films. The *x*-axis corresponds to the model output (higher means improving the stability, and vice versa).

Figure A20 (Newly added Figure S7 in SI). The feature importance rank generated using SHAP and linear regression (LR, *normalized*) for the molecular properties determining the time-intercept/ degradation onset of the capped films. The *x*-axis corresponds to the model output (higher means improving the stability, and vice versa).

Figure A21 (Newly added Figure S14 in SI). The feature importance rank generated using SHAP and multi-layer perceptron neural network (NN, *normalized*) for the molecular properties determining the time-intercept/ degradation onset of the capped films. The *x*-axis corresponds to the model output (higher means improving the stability, and vice versa).

Figure A22 (Newly added Figure S15 in SI). The feature importance rank generated using SHAP and support vector machine regression (SVR, *normalized*) for the molecular properties determining the time-intercept/ degradation onset of the capped films. The *x*-axis corresponds to the model output (higher means improving the stability, and vice versa).

Figure A23 (Newly added Figure S17 in SI). The feature importance rank generated using SHAP and gradient boosting with decision trees (**GB**, *non-normalized*) for the molecular properties determining the time-intercept/ degradation onset of the capped films. The *x*-axis corresponds to the model output (higher means improving the stability, and vice versa).

Figure A24 (Newly added Figure S18 in SI). The feature importance rank generated using SHAP and gradient boosting with decision trees (**GB**, *non-normalized*) for the molecular properties determining the time-intercept/ degradation onset of the capped films, and excluding the top stable capping layer material, PTEAI. The *x*-axis corresponds to the model output (higher means improving the stability, and vice versa).

Appendix II

The degradation of perovskite thin-films was tracked over time by photographing, that is capable of detecting degradation mechanisms changing the color of the film, in this case MAPbI₃ decomposition to MAI and PbI₂. A good approximation of the color of the samples is obtained with an RGB camera. In order to retrieve reproducible and repeatable quantitative color data, the setup was designed to keep the illumination conditions as stable as possible over time and the pictures were color calibrated.

ThorLabs DCC1645C camera (with removed IR filter to collect signal from a wider spectrum) with ThorLabs MVL6WA lens was used for taking BMP format in-situ RGB pictures of the samples every 3 minutes during the aging test. Camera settings were kept fixed in this study and they were chosen to give as light pictures as possible without oversaturating the lightest color patch in our reference color chart (which would distort the color calibration procedure). All the samples were captured simultaneously in each picture, therefore we did not have to move the sample holder. It was painted with medium gray color to prevent disturbance from over- or undersaturation of the background of the samples, as well as designed by shape to minimize reflections from the samples to the camera. The degradation chamber was covered by light-blocking curtains to remove stray light and illuminated using Advanced Illumination LED lamp and controller. We kept a miniscale reference color chart in the picture area during the aging tests, and by checking its color over time we were able to confirm illumination remained constant during the aging test.

The mean color calibrated color of each sample was determined from the photographs. In this work, X-Rite Color Checker Passport with 28 reference color patches was photographed at the beginning of each aging test in the aging chamber. The color data retrieved from the color patches was used for transforming the photographs of the samples to a stable reference color space (standard illuminant D50, standard observer CIE 1931 2 degrees). This way, the colors are comparable even though the pictures would be taken under different illumination conditions or with different camera-lens setups.

The approach chosen in our setup is to transform the samples into a larger L*a*b color space and to apply 3D thin-plate spline color warping that has been shown to be among the most accurate color warping methods for color calibration.¹⁷ Distortion between the colors of the reference color chart in real and reference color spaces is defined as:

$$D = \frac{\begin{bmatrix} V \\ O(4,3) \end{bmatrix}}{\begin{bmatrix} K & P \\ P^T & O(4,4) \end{bmatrix}},$$

where matrix V represents the colors of the reference color chart in the reference space (obtained from the color chart manufacturer), matrix P represents them in the original space, and matrix K is the distortion between the colors in the reference color chart.¹⁷ Applying the same distortion D to the samples completes the color warping:

$$\begin{bmatrix} V_s \\ O(4,3) \end{bmatrix} = \begin{bmatrix} K_{sr} & P_s \\ P_s^T & O(4,4) \end{bmatrix} D,$$

where matrices V_s and P_s represent the colors of the samples in the reference and original space, respectively, and matrix K_{sr} represents the distortion between the colors of the samples and reference color chart. After color calibration, the sample colors were transformed back to RGB space.

Finally, the “red onset” or degradation onset (Figure A5) was calculated as time-intercept ($t_{intercept}$) of the extrapolation from red channel sharpest change. The onset equation can be defined as $y = At + B$, where $A = \left(\frac{dR}{dt}\right)_{max}$, t is degradation time, and R is the calibrated red channel curve. Therefore, $t_{intercept} = -\frac{B}{A}$, where $y = 0$.

REVIEWERS' COMMENTS:

Reviewer#1

The authors have addressed all the comments and the revised manuscript may be acceptable after addressing the following minor points.

The authors have specified the situation using compressed.jpeg files, but why they have not used compressed files for all cases?

The link to the dataset and models (<https://github.mit.edu/nhartono/capping-layer>) leads to a closed internal portal that requires authorization. This is not available for outside users without the MIT account. Therefore, we recommend using an open repository e.g. GitHub, or add it into the supplementary section of the paper.

Figure S5 & Table S2 have an error. Cross-validated RMSE value is normalized case is 3.04E13. It is E11 times higher than the non-normalized case and so huge difference is only here. The authors have to specify this situation or if it's a mistake fix it.

The figure S5 is out of scale (cross-validated RMSE is readable like 300, not 3.04E13)

Tables S2, S3 & S4: please specify unit 'mins.?'

Table S4: cases of BzABr, OABr, PhABr, and PhAI are an order of magnitude smaller than other statistically significant cases. Is it possible to increase their number and/or specify what the reason for this difference is?

Reviewer#3

Reviewer#3 also commented on the revised version of the manuscript in the space dedicated to the "Remarks to the Editor" not raising any additional point.

Reviewer#4

The authors have provided satisfactory answers to my questions and I am now happy with the technical validity of the paper.

One important point should be raised though: The ML models and data have not been made available as per reviewer 1's request. The repository provided is behind a security wall requiring MIT account to login. Until this is rectified the paper remains non-reproducible.

Same as above for the accelerated test data - although this is not as critical.

** See Nature Research's author and referees' website at www.nature.com/authors for information about policies, services and author benefits

Print Email

REVIEWERS' COMMENTS

Reviewer#1

The authors have addressed all the comments and the revised manuscript may be acceptable after addressing the following minor points.

1. The authors have specified the situation using compressed.jpeg files, but why they have not used compressed files for all cases?

We thank the Reviewer#1 for raising this issue. We confirm that we use compressed JPEG files for all cases for faster data processing. The Supplementary Figure 28 (shown below) only shows 30 data points for clarity, to see the red, green, and blue (RGB) channel average value differences for different samples. The Supplementary Figure 28 shows that there is negligible difference between the uncompressed and compressed image files (< 1), therefore, we can use the compressed JPEG files for the data analysis.

Supplementary Figure 28. The average difference of respective RGB color values between .jpg and .bmp files of the 28 random samples' aging test images, showing negligible difference (< 1).

2. The link to the dataset and models (<https://github.mit.edu/nhartono/capping-layer>) leads to a closed internal portal that requires authorization. This is not available for outside users without the MIT account. Therefore, we recommend using an open repository e.g. GitHub, or add it into the supplementary section of the paper.

We apologize for this issue. We have updated our repository to be available publicly, and now it can be accessed through this link: <https://github.com/PV-Lab/capping-layer>.

3. Figure S5 & Table S2 have an error. Cross-validated RMSE value is normalized case is

3.04E13. It is E11 times higher than the non-normalized case and so huge difference is only here. The authors have to specify this situation or if it's a mistake fix it.

We thank the Reviewer#1 for clarifying this important point. We have updated the Supplementary Figure 5 to depict this. The normalized input linear regression RMSE indeed reaches 3.04E13 (Supplementary Figure 5a), which is really high. This is one of the reasons why our study focuses on non-normalized input (Supplementary Figure 5c) and trains the models based on this input for the SHAP analysis.

Supplementary Figure 5. Cross-validated RMSE for normalized input (**a**) and its inset (**b**), and non-normalized input (**c**). Error bars show the standard deviation of the cross-validated RMSE results.

4. The figure S5 is out of scale (cross-validated RMSE is readable like 300, not 3.04E13)
Tables S2, S3 & S4: please specify unit 'mins.'?

We have addressed this issue on Question 3 above and updated the Supplementary Figure 5. We also have updated Supplementary Tables 2, 3, and 4, by replacing 'mins.' with 'minutes'. Since the degradation onset unit is in minutes, the root mean square error (RMSE), will also have 'minutes' as its unit.

5. Table S4: cases of BzABr, OABr, PhABr, and PhAI are an order of magnitude smaller than other statistically significant cases. Is it possible to increase their number and/or specify what the reason for this difference is?

We thank the Reviewer#1 for asking this issue. The cases of BzABr, OABr, PhABr, and PhAI are indeed small in comparison to others, however, we have also provided Supplementary Figure 8, which highlights the ANCOVA statistical analysis for the onset results. We do not draw any conclusion for BzABr, OABr, PhABr, and PhAI samples, since they cannot be differentiated from the other sets of samples.

We do not take as many samples for fast-degrading capping layer materials. We test more slow-degrading capping layer materials to ensure the phenomenon we observe is real. In the future, we definitely would like to test this group of materials more.

Reviewer#3

Reviewer#3 also commented on the revised version of the manuscript in the space dedicated to the “Remarks to the Editor” not raising any additional point.

We thank the Reviewer#3 for all the feedbacks during the review process.

Reviewer#4

1. The authors have provided satisfactory answers to my questions and I am now happy with the technical validity of the paper.

We thank Reviewer#4 for all the feedbacks during the review process.

2. One important point should be raised though: The ML models and data have not been made available as per reviewer 1's request. The repository provided is behind a security wall requiring MIT account to login. Until this is rectified the paper remains non-reproducible. Same as above for the accelerated test data - although this is not as critical.

We apologize for this issue, it also has come to our attention that the repository is inaccessible, and we have put it here: <https://github.com/PV-Lab/capping-layer>. The dataset is also available there.